# Genetic stability of *Mycobacterium smegmatis* under the stress of first-line antitubercular agents

**Dániel Molnár[1,2†], Éva Viola Surányi[1†], Tamás Trombitás[1], Dóra Füzesi[1,2], Rita Hirmondó[1\*], Judit Toth[1,3\*]**

[1]Institute of Molecular Life Sciences, HUN-REN Research Centre for Natural Sciences, Budapest, Hungary; [2]Doctoral School of Biology and Institute of Biology, ELTE Eötvös Loránd University, Budapest, Hungary; [3]Department of Applied Biotechnology and Food Science, Budapest University of Technology and Economics, Budapest, Hungary

## eLife Assessment

This **useful** study reports on the impact of antibiotic pressure on the genomic stability of the mc2155 strain of Mycobacterium smegmatis, a model for Mycobacterium tuberculosis. The findings of the study indicate that exposure to antibiotics did not lead to the development of new adaptive mutations in controlled laboratory environments, challenging the notion that antibiotic resistance arises from drug-induced microevolution. The genomic analysis provides detailed insights into the stability of M. smegmatis following exposure to standard TB treatment antibiotics, and the evidence suggesting that antibiotic pressure does not contribute to the emergence of new adaptive mutations is **solid**.

**\*For correspondence:**
hirmondo.rita@ttk.hu (RH);
toth.judit@ttk.hu (JT)

[†]These authors contributed equally to this work

**Competing interest:** The authors declare that no competing interests exist.

**Abstract** The sustained success of *Mycobacterium tuberculosis* as a pathogen arises from its ability to persist within macrophages for extended periods and its limited responsiveness to antibiotics. Furthermore, the high incidence of resistance to the few available antituberculosis drugs is a significant concern, especially since the driving forces of the emergence of drug resistance are not clear. Drug-resistant strains of *Mycobacterium tuberculosis* can emerge through de novo mutations, however, mycobacterial mutation rates are low. To unravel the effects of antibiotic pressure on genome stability, we determined the genetic variability, phenotypic tolerance, DNA repair system activation, and dNTP pool upon treatment with current antibiotics using *Mycobacterium smegmatis*. Whole-genome sequencing revealed no significant increase in mutation rates after prolonged exposure to first-line antibiotics. However, the phenotypic fluctuation assay indicated rapid adaptation to antibiotics mediated by non-genetic factors. The upregulation of DNA repair genes, measured using qPCR, suggests that genomic integrity may be maintained through the activation of specific DNA repair pathways. Our results, indicating that antibiotic exposure does not result in de novo adaptive mutagenesis under laboratory conditions, do not lend support to the model suggesting antibiotic resistance development through drug pressure-induced microevolution.

## Introduction

Tuberculosis (TB) continues to be the most challenging, constantly present infectious disease worldwide, with 7.5 million newly reported cases and 1.3 million deaths per year (***World Health Organization, 2021***). The resurgence of TB due to the SARS-CoV-2 pandemic (***World Health Organization,***

*2023*) underscores the interconnected nature of global health and economic issues with TB incidence and control.

The causative agents of TB are members of the *Mycobacterium tuberculosis* (*M. tuberculosis*) complex. These obligate pathogen bacteria can incur a sustained threat to humanity thanks to their long-term latency (*Ehrt and Schnappinger, 2009*) and their highly unresponsive nature to antibiotics (*Hett and Rubin, 2008*; *Jankute et al., 2015*). Understanding the treatment evasion mechanisms and the outstanding stress tolerance of mycobacteria are in the spotlight of TB research (*Stallings and Glickman, 2010*; *Miggiano et al., 2020*). Drug tolerance arises when certain bacterial populations are temporarily able to survive antibiotic pressure in the absence of drug resistance-conferring mutations. Upon exposure to bactericidal drugs, tolerant mycobacteria are eliminated at a lower rate than the fully susceptible population (*Balaban et al., 2019*). Several interconnected biological pathways are involved in the emergence and establishment of a drug-tolerant state (*Boshoff et al., 2004*; *Walter et al., 2015*) including metabolic slowdown, metabolic shifting, cell wall thickening, and transcriptional regulation-guided adaptation (*Goossens et al., 2020*). For example, several efflux pumps are upregulated under antibiotic stress (*Louw et al., 2011*; *Wiuff et al., 2005*). In addition to temporary drug tolerance, the occurrence of genotypic resistance against the few useable antituberculotics is also recurrent (*Dookie et al., 2018*). Interestingly, horizontal gene transfer, which is a major contributor to antibiotic resistance in other species does not appear to function in members of the *M. tuberculosis* complex (*Gray and Derbyshire, 2018*; *Madacki et al., 2021*). Therefore, any resistant genotype can only emerge by de novo mutagenesis.

It is now commonly accepted that the *M. tuberculosis* population within individual TB patients can be more heterogeneous than was traditionally thought (*Hingley-Wilson et al., 2013*; *Liu et al., 2015*). The coexistence of both drug-resistant and drug-sensitive strains in a single patient, or even several drug-resistant strains with discrete drug resistance-conferring mutations has been described in clinical isolates (*Asare-Baah et al., 2021*; *Lozano et al., 2021*; *Pérez-Lago et al., 2016*). Warren et al. found that the occurrence of mixed infections reached 19% of the examined patients in South Africa by using a PCR-based strain classification method (*Warren et al., 2004*). Mixed infections can result from (i) simultaneously or sequentially acquired infections by different strains or (ii) genomic evolution of a strain under mutagenic pressure within the host (termed microevolution) and consequent coexistence of several populations. Accordingly, the emergence of genetically encoded resistance may either be due to microevolution or to the spreading of already existing variants from polyclonal infections under drug pressure. The difference between these two underlying mechanisms for the emergence of drug resistance is highly relevant to the treatment of TB. The investigation of stress-induced mutagenesis in mycobacteria has been based on fluctuation assays (*Ford et al., 2013*; *Gillespie et al., 2005*) besides several indirect evidence from descriptive studies (*Al-Hajoj et al., 2010*; *Navarro et al., 2017*; *Herranz et al., 2018*; *Ley et al., 2019*; *Sun et al., 2012*). However, we propose that combining mutation accumulation assays, analyzed through whole-genome sequencing, with phenotypic fluctuation assays is essential for identifying the source of the antibiotic resistance phenotype. Some studies demonstrate the simultaneous presence of several subpopulations within the same host which they interpret as an indication of being prone to microevolution (*Navarro et al., 2011*; *Pérez-Lago et al., 2016*). It is also possible that certain strains have intrinsically higher mutability. For example, the lineage 2 strains of the Beijing genotype exhibited a higher mutation rate (*Ford et al., 2011*). On the other hand, others found stable *M. tuberculosis* genomes with no or only a few emerging genomic changes over prolonged periods of treatment (*Herranz et al., 2018*). Genotyping has enabled researchers to describe cases of co-infection by ≥2 different strains (mixed infection) or the coexistence of clonal variants of the same strain (*Muwonge et al., 2013*; *Navarro et al., 2011*; *Shamputa et al., 2006*). Introducing whole genome sequencing into this field still leaves the distinction between mixed infections with multiple similar strains and strains that have arisen by microevolution elusive. Depending on the elapsed time between two sample collections, the stepwise acquisition of mutations might be missed, and the observed diversity may reflect concurrently existing subclones rather than newly emerged mutations (*Ley et al., 2019*). In addition, a single sputum sample usually does not represent the whole genomic diversity of the infection (*Liu et al., 2015*; *Shamputa et al., 2006*). Cell culturing can also lead to additional artefacts (*Doyle et al., 2018*; *Metcalfe et al., 2017*). The lack of standardized reporting of genome sequencing analyses also limits our ability to draw conclusions on within-host microevolution (*Ley et al., 2019*). Therefore, although

several factors such as drug pressure and disease severity have been suggested to drive within-host microevolution and diversity (*O'Neill et al., 2015*; *Trauner et al., 2017*) and it is now accepted that the *M. tuberculosis* population within individual patients can be heterogeneous, we could not find any unequivocal proof for explaining the mechanism of emergence of the observed genomic diversity which gives rise to drug resistance.

Therefore, to advance our knowledge on the effect of antibiotics on mycobacterial mutability, we conducted experiments under controlled laboratory conditions. We used *Mycobacterium smegmatis* (*M. smegmatis*) for our investigations. This non-pathogenic relative of the medically relevant *Mycobacterium* species shares most DNA metabolic pathways with the medically relevant strains. Davis and Forse compared the sequences of proteins involved in base excision repair and nucleotide excision repair pathways in *E. coli* and their homologs in *M. smegmatis* and *M. tuberculosis* and found that there is a high degree of conservation between the DNA repair enzymes in *M. smegmatis* and *M. tuberculosis* (*Davis and Forse, 2009*; *Kurthkoti and Varshney, 2012*). Bioinformatic analyses of completely sequenced mycobacterial genomes, including *M. tuberculosis* (*Camus et al., 2002*), *M. leprae* (*Silva et al., 2022*), *M. bovis* (*Garnier et al., 2003*; *Zimpel et al., 2017*), *M. avium, M. paratuberculosis,* and *M. smegmatis* (*Mohan et al., 2015*) also demonstrated through the comparison of genes participating in many of the DNA repair/recombination pathways that the basic strategy used to repair DNA lesions is conserved (*Singh, 2017*; *Singh et al., 2010*). Durbach et.al, investigated mycobacterial SOS response and showed that the *M. tuberculosis, M. smegmatis,* and *M. leprae* LexA proteins are functionally conserved at the level of DNA binding (*Durbach et al., 1997*). In our earlier paper, we also compared the enzymes of thymidylate biosynthesis in *M. tuberculosis* and *M. smegmatis* and found high conservation (*Pecsi et al., 2012*). Considering *M. smegmatis* is non-pathogenic and fast-growing, it provides an attainable model to obtain information on genomic changes under drug pressure in *M. tuberculosis*.

We systematically investigated the effects of currently used TB drugs on genome stability, tolerance/ resistance acquisition, activation of the DNA repair system, and the cellular dNTP pool. We focused particularly on drugs used in the standard treatment of drug-susceptible TB, comprising isoniazid (INH), rifampicin (RIF), ethambutol (EMB), and pyrazinamide (PZA), the so-called first-line antibiotics (*Grace et al., 2019*). We also used a second-line antibiotic, ciprofloxacin (CIP). We found that following exposure to these antibiotics, the activation of DNA repair pathways maintains genomic integrity, while non-genetic factors convey quick adaptation to stress conditions. Notably, even with prolonged antibiotic exposure exceeding 230 bacterial generations, we observed no significant increase in the mutation rate, suggesting the absence of de novo adaptive mutagenesis.

**Table 1.** Summary of the applied drug treatments and their phenotypic consequences.

| Treatment | | | | Liquid culture experiments | | | | Agar plate experiments | |
|---|---|---|---|---|---|---|---|---|---|
| Category | Long name | Abbreviation | Mechanism of action | Subinhinitory concentration | CFU compared to control | Cell length [μm] | Cell width [μm] | Subinhinitory concentration | CFU compared to control |
| | Isoniazid | INH | Cell wall synthesis inhibitor | 150 μg/ml | 80% | 1.8±0.5 | 0.41±0.07 | 2 μg/ml | 2.2 % |
| | Ethambutol | EMB | | 100 μg/ml | 70% | 2.0±0.8 | 0.55±0.17 | 0.2 μg/ml | 10.5 % |
| | Rifampicin | RIF | RNA synthesis inhibitor | 3 μg/ml | 60% | 6.6±2.4 | 0.68±0.09 | 25 μg/ml | 0.00052 % |
| First line antibiotics | Combination treatment | COMBO | WHO first line therapy | 10 μg/ml PZA, 15 μg/mL INH, 10 μg/ml EMB, 0.3 μg/mL RIF | 6% | 2.8±0.7 | 0.47±005 | 1 μg/ml PZA, 0.2 μg/mL INH, 0.02 μg/mL EMB, 2.5 μg/mL RIF | 0.39 % |
| Second line antibiotics | Ciprofloxacin | CIP | Gyrase inhibitor | 0.3 μg/ml | 20% | 11.1±4.0 | 0.59±0.1 | 0.3 μg/ml | 0.00018 % |
| | Mitomycin-C | MMC | DNA alkylation | 0.01 μg/ml | 20% | 9.8±4.6 | 0.68±0.11 | 0.0005 μg/ml | 0.96 % |
| DNA damage controls | Ultraviolet radiation | UV | Pyr dimers, DSBs | ND | ND | ND | ND | 150 J/m$^2$ | 11 % |
| N/A | Non-treated | Mock | N/A | N/A | 100% | 2.8±0.9 | 0.44±0.08 | N/A | 100 % |

## Results

### Adapting stress conditions and assessing their impact on cell viability and morphology

For an efficient TB treatment, first-line antituberculotics are used in combination in the clinics (isoniazid – INH; ethambutol – EMB; rifampicin – RIF; pyrazinamide - PZA) (*Trauner et al., 2017*). To model this drug pressure in our study, we also combined the four first-line drugs in addition to applying them one by one. We added a second-line antibiotic, CIP. MitomycinC (MMC) and ultraviolet (UV) irradiation were used as positive controls for direct DNA damage (*Crowley et al., 2006*; *O'Sullivan et al., 2008*). We optimized the drug concentration for all applied treatments. First- and second-line antituberculosis drugs were used in sublethal concentrations to convey a measurable phenotypic effect while allowing to keep an adequate number of cells for the MA experiments on a plate and for the downstream measurements in liquid culture (*Figure 1—figure supplement 1*; *Figure 1—figure supplement 2* and *Table 1*). In the first-line combination treatment, a 10-fold reduced concentration

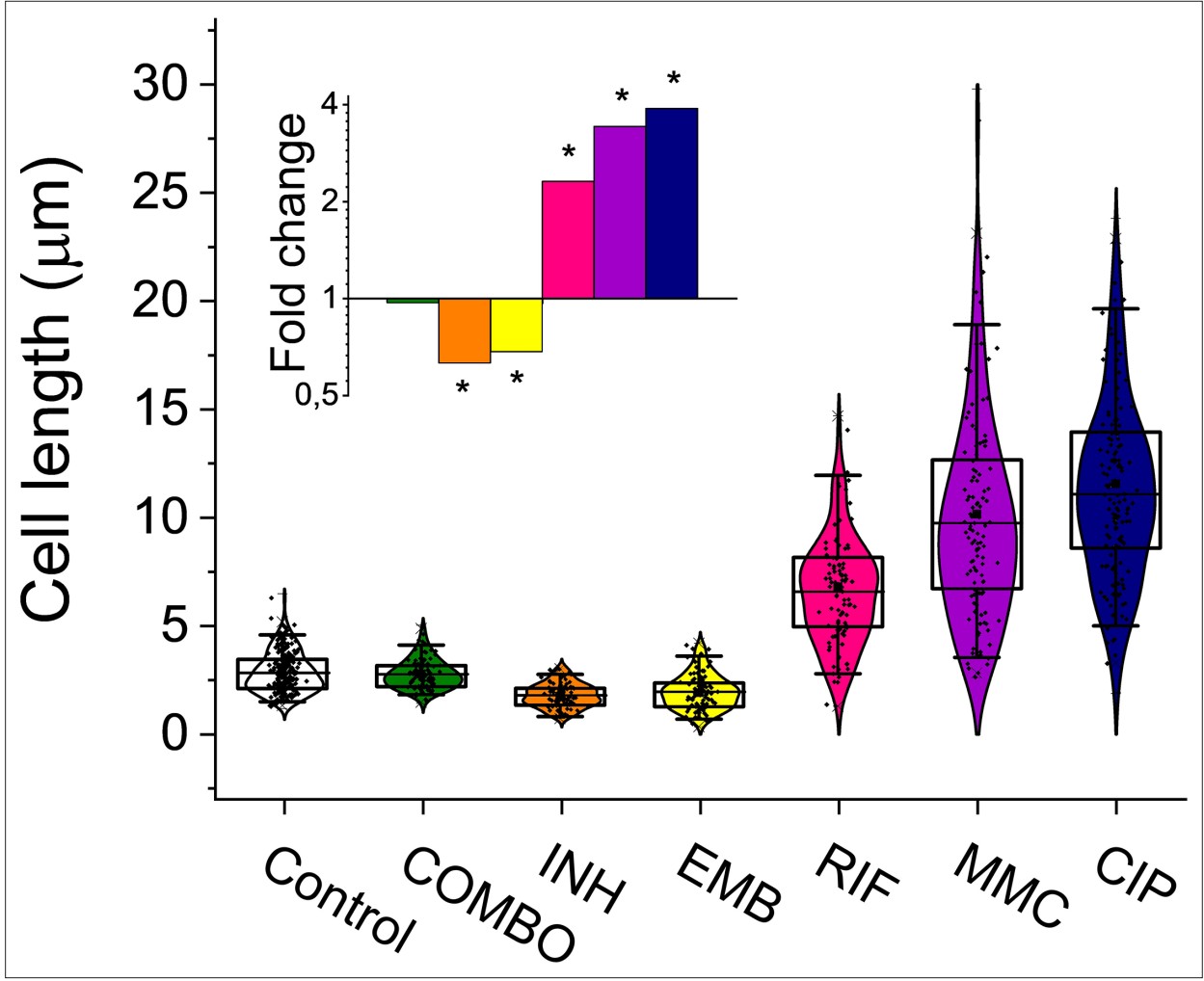

**Figure 1.** Cell length distribution of *M.smegmatis* cells treated with different drugs. Horizontal lines represent the mean of the plotted data points (n=84–212). The inset shows the fold changes in cell length compared to the untreated control on a log2 axis, highlighting the phenotypic effect of each treatment. * indicates data significantly different from the control at p=0.0001. Numerical values and additional statistical parameters are provided in *Figure 1—source data 1*.

The online version of this article includes the following source data and figure supplement(s) for figure 1:

**Source data 1.** Cell dimensions of *M. smegmatis* treated with different drugs.

**Figure supplement 1.** Treatment optimization in liquid culture.

**Figure supplement 2.** Treatment optimization on agar plates.

of each separately adjusted drug had to be applied in both liquid and agar media to allow the survival of enough cells for the analyses (*Table 1*). The fact that a lower dose of antibiotics applied in combination resulted in higher CFU reduction indicates the synergistic effect of the first-line antibiotics on *M. smegmatis* growth inhibition (*Table 1*). After an 8 hr drug treatment, we determined the viable cell count by CFU measurements (*Table 1*). The bacteriostatic drugs INH and EMB caused moderate CFU decrease in liquid cultures compared to the control (*Table 1*, *Figure 1—figure supplement 1*), consistent with their mechanism of action (*Alland et al., 2000*). To quantify the phenotypic effect of the applied drug treatments in liquid cultures, we analyzed the cellular dimensions using microscopy (*Figure 1* and *Table 1*). The observed morphological changes provided evidence of the treatments' effectiveness (*Figure 1* and *Table 1*). Specifically, following RIF, CIP, and MMC treatments, we observed cells elongating by more than twofold, whereas INH and EMB treatments led to a reduction in cell length. The combination treatment did not affect the cell size (*Figure 1* and *Table 1*).

We also assayed the clinically relevant drug PZA. However, *M. smegmatis* was reported to exhibit an intrinsic resistance to PZA (*Zhang et al., 1999*). Indeed, PZA treatment alone, even at high concentrations in acidic conditions, did not affect cell viability in our experiments (*Figure 1—figure supplement 1*). Regardless of its inefficacy as a monotherapy, we included PZA in the combination treatment, as we could not rule out the possibility that PZA interacts with the other three drugs or that PZA elimination mechanisms are equally active in *M. smegmatis* under this regimen.

## The genome of *M. smegmatis* remains stable even under antibiotic pressure

16 independent *M. smegmatis* MC$^2$ 155 lineages for each stress treatment condition and 56 lineages for the mock control were initiated and cultured from single colonies. The stress-treated lines and some of the mock lines were maintained through 60 days on agar plates. The rest of the mock lines were maintained through 120 days on agar plates. Drug-treated lineages were maintained for shorter times as more mutations were expected to arise under drug pressure. We measured an average generation time of 6.3±0.35 hr on the plate within the timeframe of a single passage. Therefore, bacteria produced on average 230 generations during the 60 day treatment. Following the treatment on solid plates, we expanded each lineage in a liquid culture without drug pressure and isolated genomic DNA. All lineages were sent to WGS to reveal the mutational events induced by the drug treatments. We set conditions to obtain at least 30–60 x sequencing depth for all positions per independent lineage. The ancestor colony was also sent for sequencing to detect already existing variations compared to the reference genome. According to the WGS results, our *M. smegmatis* ancestor strain carried 151 various mutations compared to the *M. smegmatis* reference genome deposited in the GenBank. These mutation hits were also found in all treated and untreated lineages and were omitted from further calculations as these are specific variations of our laboratory strain. We also removed those mutation hits that were found in any other independent lineage at the same position in any depth.

A surprisingly few new mutations were detected after carefully cross-checking the sequencing data. We found that a maximum of one mutation per lineage occurred during the 60 day drug treatments. Also, a maximum of one mutation per lineage was detected during the 60- or 120 day mock treatment (16 newly generated mutations for 56 lineages). We calculated a $1×10^{-10}$ mutation rate for our untreated *M. smegmatis mc$^2$155* strain. To our great surprise, the mutation rates of all treated lineages fell in one order of magnitude ($4×10^{-11}$ - $3×10^{-10}$) except for the UV treatment used for positive control (*Figure 2B*).

We analyzed each mutation except those obtained following the UV treatment and found no sign of adaptive changes (*Table 2*). The Excel file containing the positions of all obtained mutations, including those of the UV sample, is provided in the archive deposited for the article (https://doi.org/10.6084/m9.figshare.25028186).

We assessed the drug sensitivity of the MA strains by measuring the MIC of each drug on three randomly selected strains from both the mock-treated and stressed MA groups. Contrary to the mutation rate results obtained from genomic sequencing data, the MIC values for the MA strains were higher than those of the mock-treated strains (comparable data in line with *Nyinoh, 2019*), indicating

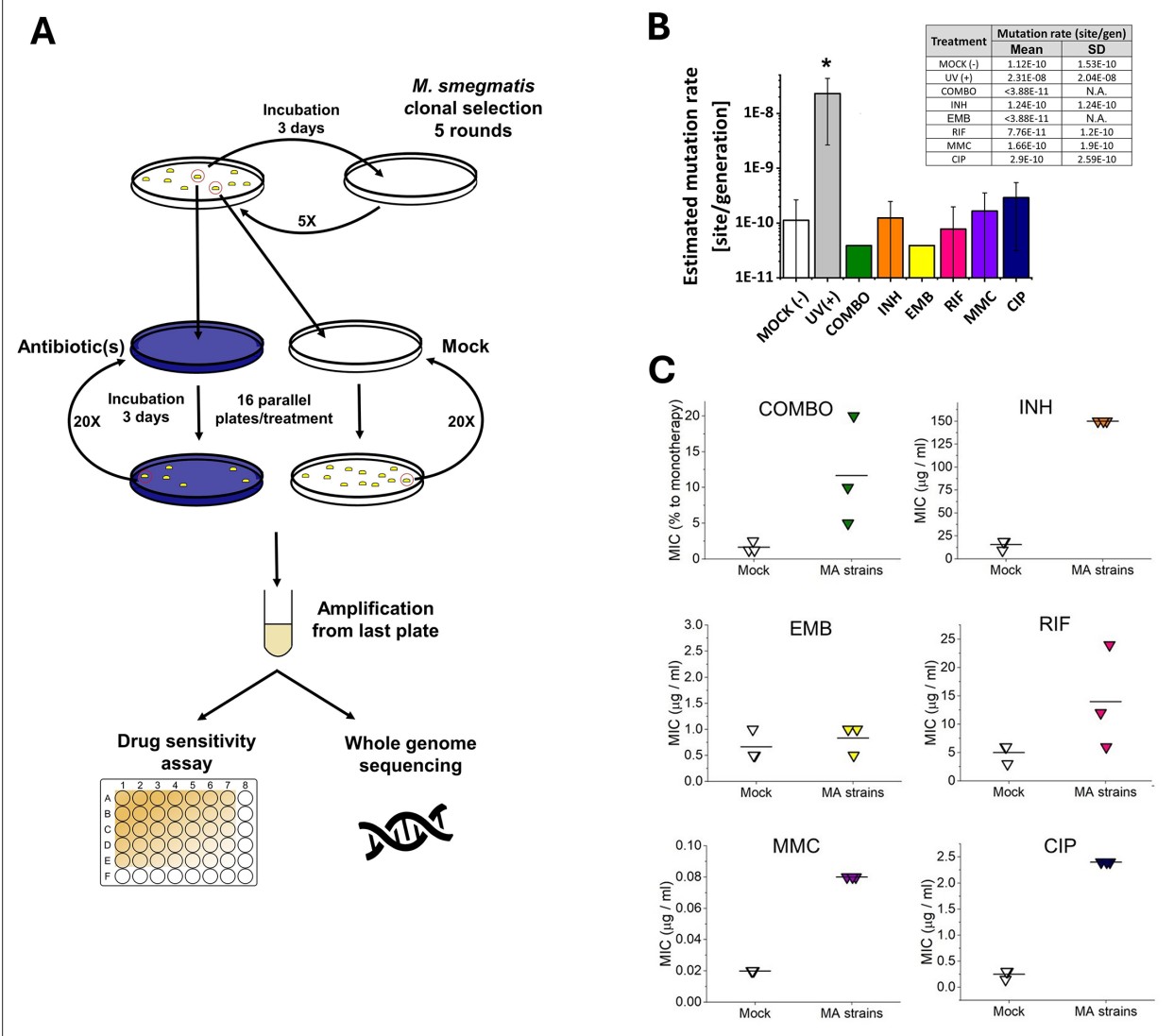

**Figure 2.** Mutation accumulation (MA) experiment and the resulting genotypic and phenotypic changes in wild-type *M. smegmatis mc²155* strains under antibiotic pressure. (**A**) Experimental design. (**B**) Mutation rates determined through genome sequencing of the drug-treated cells as an output of the MA process. UV(+) serves as a control reference for DNA damage. Columns represent averages, and error bars indicate the standard deviations of three individually sequenced samples. Statistical significance is marked by an asterisk (*), with a p-value of 0.05. For numerical data see *Figure 2—source data 1*. (**C**) Phenotypic drug sensitivity in drug-treated strains. Three individual minimal inhibitory concentration (MIC) determinations are presented, with the mean indicated by a horizontal line. For numerical data see *Figure 2—source data 2*.

The online version of this article includes the following source data for figure 2:

**Source data 1.** Numerical data for mutation rates of wild-type *M. smegmatis* mc2155 strains under antibiotic pressure.

**Source data 2.** Phenotypic drug sensitivity (MIC) in drug- treated strains.

phenotypic adaptation to the applied drugs (*Figure 2C*, *Figure 2—source data 2*). However, for the EMB treatment, we observed no increase in MIC, despite repeating the experiment several times.

## The DNA repair system shows a treatment-specific activation pattern

To investigate a possible reason for detecting so few newly generated mutations under antibiotic pressure, we studied whether the DNA repair pathways and other elements of the stress response potentially involved (*Romero et al., 2011*) were activated under drug pressure. The mycobacterial DNA repair system is highly redundant, many of its enzymes have overlapping functions (*Malshetty et al., 2010*; *Singh, 2017*; *Srinath et al., 2007*). Although canonical mismatch repair proteins are thought to be missing, a recently described protein, NucS is encoded with a similar function

**Table 2.** Analysis of the genomic changes detected in the mutation accumulation experiment using whole genome sequencing.

| Chromosome position | Sample | Reference | Mutation | AA mutation | Gene code | UniProt protein name | Gene ontology (GO) | Experiment |
|---|---|---|---|---|---|---|---|---|
| 5214897 | cip_b | A | AG | Leu87 frameshift 148stop | MSMEG_5116 | Uncharacterized protein | N/A | Mutation accumulation (MA) |
| 3614832 | cip_b | C | T | Pro139Leu | MSMEG_3554 | N5,N10-methylene-tetrahydromethanopterin reductase | xidoreductase activity, acting on paired donors, with incorporation or reduction of molecular oxygen [GO:0016705] | |
| 2208516 | cip_b | G | GA | Leu282 frameshift 283stop | MSMEG_2133 | Uncharacterized protein | N/A | |
| 5861538 | cip_c | G | GC | Leu168 frameshift | MSMEG_5792 | UPF0678 fatty acid-binding protein-like protein MSMEG_5792/MSMEI_5639 | intracellular transport [GO:0046907] | |
| 3415264 | cip_c | T | TC | Leu206 frameshift 257stop | MSMEG_3338 | Oxidoreductase, FAD/FMN-binding | FMN binding [GO:0010181]; oxidoreductase activity [GO:0016491] | |
| 2033295 | cip_c | A | AG | Leu72 frameshift 258stop | MSMEG_1954 | ABC1 family protein | N/A | |
| 1988098 | cip_c | A | AG | N/A | Intergenic region | intergenic | N/A | |
| 1533730 | inh_b | C | CTCG | Asp201_INSERTION | MSMEG_1431 | Cytochrome P450-terp (EC 1.14.-.-) | heme binding [GO:0020037]; iron ion binding [GO:0005506]; monooxygenase activity [GO:0004497]; oxidoreductase activity, acting on paired donors, with incorporation or reduction of molecular oxygen [GO:0016705] | |
| 994997 | inh_C | G | A | N/A | intergenic | N/A | N/A | |
| 5777585 | inh_C | C | T | Val99Met | MSMEG_5688 | Regulatory protein, MarR | GO:0003700 DNA-binding transcription factor activity; GO:0006355 regulation of DNA-templated transcription | |
| 1508883 | mmc_a | C | G | Ala300Ala (neutral) | MSMEG_1407 | N/A | N/A | |
| 4598387 | mmc_a | C | G | Ala371Arg | MSMEG_4513 | Polyketide synthase | transferase activity, transferring acyl groups [GO:0016746] | |
| 6786854 | mmc_a | G | A | Trp104stop | MSMEG_6740 | 1-aminocyclopropane-1-carboxylate deaminase (EC 3.5.99.7) | 1-aminocyclopropane-1-carboxylate deaminase activity [GO:0008660]; pyridoxal phosphate binding [GO:0030170]; amine catabolic process [GO:0009310] | |
| 5313643 | mmc_c | C | T | N/A | intergenic | N/A | N/A | |
| 1865825 | mock_b | G | GC | Ala351 frameshift | MSMEG_1780 | Natural resistance-associated macrophage protein | metal ion transmembrane transporter activity; metal ion transport; membrane; | |
| 3722101 | mock_b | A | C | Asn185Thr | MSMEG_3656 | ABC transporter, permease/ATP-binding protein | | |
| 58213 | mock_c | T | TC | N/A | MSMEG_0037 | tRNA-Leu | N/A | |
| 4104684 | mock_c | T | C | Val70Ala | MSMEG_4033 | TetR-family protein transcriptional regulator | GO:0006350, Sequence-specific dna binding transcription factor activity, Regulation of transcription, dna-dependent | |
| 5118524 | mock_c | C | CG | Asp89 frameshift 143stop | MSMEG_5021 | Alcohol dehydrogenase, zinc-containing | Oxidoreductase activity, Zinc ion binding, Oxidation-reduction process | |
| 5217666 | mock_g | G | A | Thr200Thr (neutral) | MSMEG_5119 | L-glutamate gamma-semialdehyde dehydrogenase | Mitochondrial matrix, Oxidation-reduction process, Proline biosynthetic process, 1-pyrroline-5-carboxylate dehydrogenase activity | |
| 2970975 | mock_i | T | C | Arg155Gly | MSMEG_2908 | 2-Keto-3-deoxy-gluconate kinase | kinase activity [GO:0016301] | |
| 2970982 | mock_i | C | T | Arg153Glu | MSMEG_2908 | 2-Keto-3-deoxy-gluconate kinase | kinase activity [GO:0016301] | |

*Table 2 continued on next page*

*Table 2 continued*

| Chromosome position | Sample | Reference | Mutation | AA mutation | Gene code | UniProt protein name | Gene ontology (GO) | Experiment |
|---|---|---|---|---|---|---|---|---|
| 3306164 | mock_i | G | A | Glu1151Glu (neutral) | MSMEG_3225 | Ferredoxin-dependent glutamate synthase 1 (EC 1.4.7.1) | 3 iron, 4 sulfur cluster binding [GO:0051538]; glutamate synthase (ferredoxin) activity [GO:0016041]; metal ion binding [GO:0046872]; glutamate biosynthetic process [GO:0006537]; glutamine metabolic process [GO:0006541] | Mutation accumulation (MA) |
| 5805844 | mock_i | C | T | Val237Val (neutral) | MSMEG_5721 | Acetyl-CoA acetyltransferase | transferase activity, transferring acyl groups other than amino-acyl groups [GO:0016747] | |
| 4987517 | mock_j | G | A | Leu30Leu (neutral) | MSMEG_4890 | Alkyl hydroperoxide reductase AhpD (EC 1.11.1.28) (Alkylhydroperoxidase AhpD) | alkyl hydroperoxide reductase activity [GO:0008785]; hydroperoxide reductase activity [GO:0032843]; peroxidase activity [GO:0004601]; peroxiredoxin activity [GO:0051920]; response to oxidative stress [GO:0006979] | |
| 6406902 | mock_j | T | TG | N/A | intergenic | N/A | N/A | |
| 491016 | mock_k | C | T | N/A | intergenic | N/A | N/A | |
| 2287781 | mock_k | G | A | Gly199Asp | MSMEG_2207 | Beta-ketothiolase | transferase activity, transferring acyl groups other than amino-acyl groups [GO:0016747] | |
| 3438752 | rif_a | A | AC | Arg17 frameshift 175stop | MSMEG_3366 | Isonitrile hydratase, putative | N/A | |
| 5773058 | rif_a | C | T | Glu67Lys | MSMEG_5682 | Uncharacterized protein | integral component of membrane [GO:0016021] | |
| 6220187 | CIPB0.3 | G | T | Trp53Cys | MSMEG_6151 | Alpha/beta hydrolase fold-1 | epoxide hydrolase activity [GO:0004301] | Fluctuation assay with CIP treatment |

| | Target | INH | EMB | RIF | COMBO | CIP | MMC |
|---|---|---|---|---|---|---|---|
| **Base Excision Repair** | AlkA | 2.68 | -1.19 | 2.12 | 8.58** | -1.71** | -1.48 |
| | End | -1.22 | 1.21 | -2.05 | 2.56* | 3.69 | 2.26** |
| | Mpg | -2.91 | 1.05 | 1.37** | 5.55** | -2.68** | -1.46 |
| | MutM1 | -1.70 | -3.72* | -1.33 | 4.03** | -1.66 | -1.58 |
| | MutY | -3.04 | -1.19 | -1.12 | 2.66 | -1.39 | -1.52 |
| | Nei1 | -1.77 | -5.73 | -1.34 | 2.62 | -1.33 | -1.52 |
| | Nei2 | -4.25** | -3.28 | -1.33 | 2.03 | -1.30 | 1.37 |
| | Ogt | 1.90 | 1.84 | 3.19 | 4.98** | 1.31 | -1.39 |
| | TagA | -3.43* | -1.11 | -1.24 | 2.22* | 4.44** | 1.93 |
| | UdgB | -1.22 | 3.51 | 1.69 | 5.43** | 1.04 | -1.01 |
| | UdgX | -2.04 | -2.32* | 3.29 | 1.29 | 1.18 | -1.17 |
| | Ung | -1.46 | 2.94 | -1.01 | -1.24 | -2.68** | -1.98 |
| | XthA | -1.02 | 1.48 | 1.03 | 3.55** | 1.10 | -1.75 |
| **Nucleotide excision repair** | Mfd | -1.74 | 1.12 | 1.38 | 1.50 | 3.83** | 1.42 |
| | UvrA | -1.04 | -1.14 | 1.39 | -1.05 | 1.98* | 1.06 |
| | UvrB | -1.01 | 1.13 | 1.45 | 1.79 | 5.51** | 2.06 |
| | UvrC | -4.08** | -2.13 | -2.36 | 2.55 | -1.14 | -1.30 |
| | UvrD | -2.09** | -5.76 | -1.30 | 1.99* | 2.45** | 1.60 |
| **dNTP pool sanitization** | Dcd:dut | -1.06 | 1.27 | -1.50 | 1.97 | 1.69 | -1.07 |
| | Dut | -2.22* | -3.39 | -1.14 | 2.31 | -1.34 | -1.74 |
| | MutT1 | -3.30 | -2.16 | -1.12 | 1.86 | -1.51 | -1.15 |
| | MutT2 | 1.42 | -3.22 | -2.53 | 6.08** | -2.59** | -1.66 |
| | MutT3 | -1.34 | -4.11 | -2.55 | 1.93* | 1.06 | -1.38 |
| | MutT4 | -1.26 | 1.00 | 2.15 | 1.13 | 1.90 | -1.24 |
| | ThyA | -18.96** | -4.95 | -2.16 | 1.67 | -2.53** | -1.63 |
| | ThyX | -3.64 | -1.90 | -2.38 | 2.41 | -1.03 | -1.79 |
| **DNA synthesis** | DinB1 | 2.45 | -3.09 | -1.44 | 2.10 | -1.07 | 1.18 |
| | DinB2 | 1.10 | 1.52 | 1.38 | 3.53** | 12.69** | 7.51 |
| | DNA ligase | -1.48 | 1.70 | -1.04 | 1.85 | 3.17* | 1.99 |
| | DnaE2 | 2.65 | -1.22 | -1.52 | 5.63** | 10.40** | 8.48** |
| | PolA | -1.88 | 1.06 | 1.68 | 1.79 | 3.77** | 1.35 |
| **Double-strand break repair** | AdnA | -1.49 | 1.20 | 2.28 | 1.15 | 14.12** | 7.90* |
| | LexA | 1.12 | 2.05 | -1.18 | -1.18 | 13.34** | 6.27 |
| | RecA | -1.04 | 1.41* | -1.33 | -1.41 | 7.50** | 6.69** |
| | RecX | 1.05 | 1.66 | -1.67 | 1.32 | 6.23** | 5.34* |
| **Mismatch repair** | NucS | 1.57 | 1.13 | -1.08 | 2.89 | 1.03 | -1.11 |
| **Peroxidase** | AhpC | -1.82 | -1.76 | -4.38** | 1.26 | -1.33 | -1.4 |
| | KatG1 | -3.00** | -4.21 | -1.24 | 17.17** | 1.42 | 1.02 |

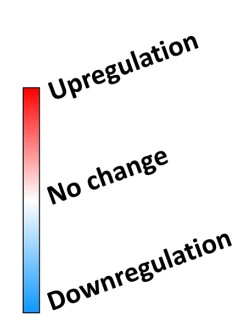

**Figure 3.** Changes in the expression of DNA repair genes upon stress treatments. Gene expression changes are normalized to the mock-treated control using the SigA and Ffh reference genes. Upregulation is numerically interpreted as fold change; downregulation is interpreted as –1/ (fold change) in the heatmap. *p<0.1; **p<0.05. For raw data see *Figure 3—source data 1*.

The online version of this article includes the following source data and figure supplement(s) for figure 3:

**Source data 1.** Numerical qPCR results.

**Figure supplement 1.** Stability analysis of reference genes using the geNorm algorithm.

**Figure supplement 2.** Specificity assessment of the employed primers.

**Figure supplement 3.** Heatmap with clustering for gene expression changes upon treatment.

(*Castañeda-García et al., 2017*). We investigated the expression pattern of DNA repair genes in all known DNA repair pathways in mycobacteria including NucS using RT-qPCR, a method suitable to accurately show changes in transcript levels. The measured relative expression levels are presented in *Figure 3*, grouped by functional relevance, with consistent heatmap coloring across all measurements.

*Figure 3—figure supplement 3* shows a clustered heatmap without prior functional grouping. Numerical data for expression level changes are provided in *Figure 3—source data 1*.

Treatments with the two antibiotics affecting cell wall synthesis (INH and EMB) show similar patterns in the expression levels with an overall downregulation of DNA repair genes. On the contrary, CIP and MMC, drugs directly targeting DNA integrity induce a pattern marked by a moderate to strong over-expression of nucleotide excision and double-strand break (DSB) repair genes, respectively (*Figure 3* and *Figure 3—figure supplement 3*). DNA polymerases DinB2 and DnaE2 involved in these DNA repair pathways are also strongly overexpressed (*Figure 3* and *Figure 3—figure supplement 3*). RIF, the DNA-dependent RNA polymerase inhibitor does not seem to induce any change in the expression pattern of the investigated genes except for the Ahp peroxiredoxin (*Figure 3* and *Figure 3—figure supplement 3*). As a result of the first line combination (COMBO) treatment, 14 out of 38 investigated genes are significantly (p<0.05) upregulated. More than fourfold upregulation can be measured for 5 members of the base excision repair pathway. In addition, the MutT2 dNTP pool sanitization enzyme and the error-prone DNA polymerases are also strongly upregulated. (*Figure 3* and *Figure 3—figure supplement 3*). Interestingly, however, the DSB repair enzymes are exempt from this overall upregulation tendency (*Figure 3* and *Figure 3—figure supplement 3*). The strongest measured effect of all is the 17-fold expression increase of the KatG1 peroxidase (*Figure 3*). When the first line antibiotics

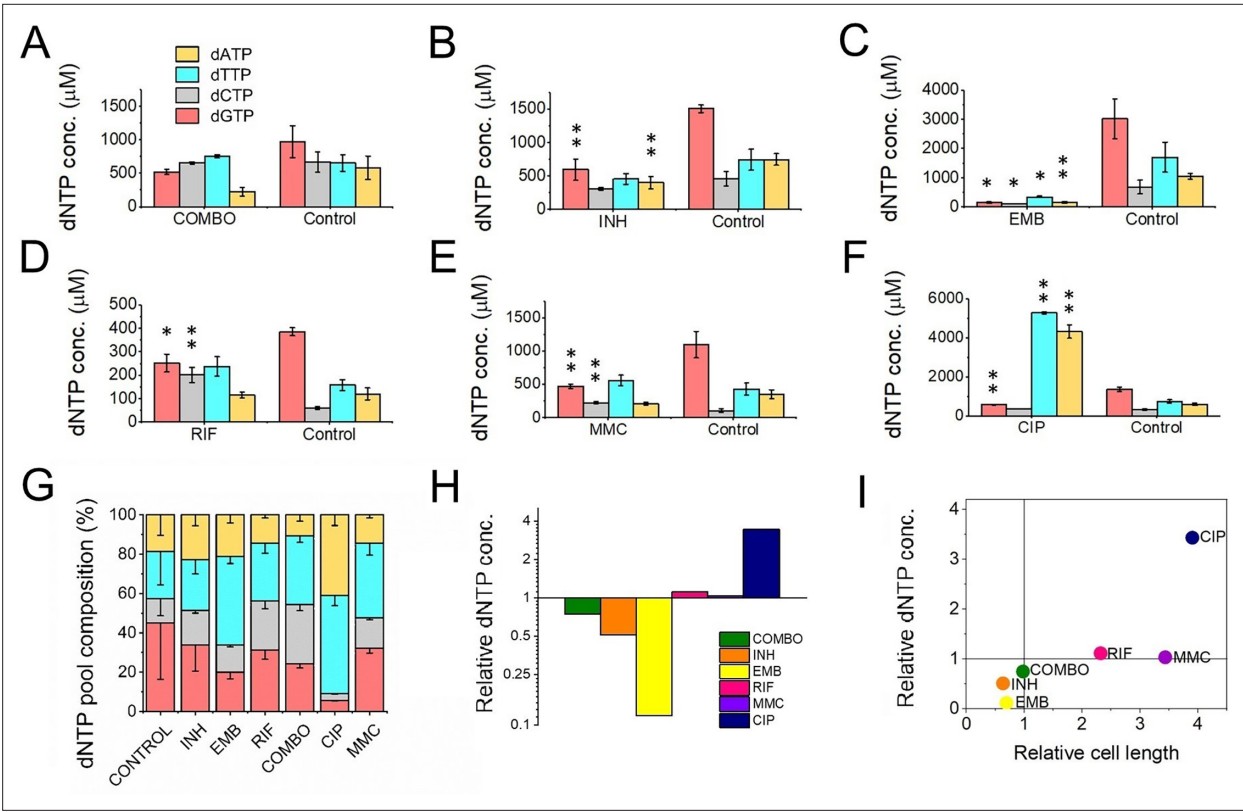

**Figure 4.** First-line antituberculotic treatments and DNA damaging agents alter dNTP concentrations in the cell. (**A–F**) Cellular dNTP concentrations in drug-treated *M. smegmatis*. dNTP levels were measured in cellular extracts and normalized to the average cell volume for each treatment, yielding the concentrations shown. Each drug treatment and dNTP quantification included a corresponding control to account for potential fluctuations in growth and experimental conditions. Note the different scales on the y-axis. Data bars represent the averages of three biological replicates each carried out in three technical replicates; error bars represent SE. The p-values from the t-tests calculated for the measured differences are provided in *Figure 1—source data 1*, with significance indicated in the figure by asterisks as follows (**) for p<0.04 and (*) for p<0.07. (**G**) dNTP pool compositions of drug-treated bacteria. The large error bars in the control data arise from the combination of individual controls measured for each treatment. (**H**) Summed molar concentration of all four dNTPs compared to the control for each treatment. The y-axis is on a log2 scale to equally represent both increases and decreases. (**I**) Correlation of relative cell size (determined from cell lengths, compared to control cells) to relative total dNTP concentration for each treatment.

The online version of this article includes the following source data for figure 4:

**Source data 1.** dNTP concentrations in cellular extracts upon treatment with drugs.

were used one by one, significant expression change could only be observed upon the INH treatment (4/38 genes) and in the opposite direction (downregulation).

## All but the combination treatment alters the size and balance of dNTP pools

It was shown that dNTP pools are crucial for genome maintenance and proper DNA synthesis (*Kumar et al., 2010*; *Mathews, 2006*; *Nordman and Wright, 2008*; *Yao et al., 2013*). Imbalanced or altered levels of dNTPs could cause an increased rate of DNA lesions and, therefore, may play a role in the development of drug resistance. Therefore, we measured cellular dNTP concentrations and ratios in the function of the applied drug treatments using a fluorescent detection-based method optimized in our lab (*Szabó et al., 2020*). We used MMC treatment as a positive control as this is a generally used positive control for DNA damage (*Kurthkoti et al., 2008*; *O'Sullivan et al., 2008*). To calculate cellular concentrations, we used the cellular volumes determined from measured cell dimensions *Figure 1—source data 1*. Interestingly, we found altered dNTP pools upon most treatments (*Figure 4* and *Figure 4—source data 1*). The CIP treatment resulted in the most remarkable differences in particular for dATP and dTTP concentrations which increased ~ sevenfold accompanied by a decrease in the dGTP concentration (*Figure 4F and H*). RIF and MMC treatments promoted an increase in the dGTP and dCTP pools (*Figure 4D–E*). The INH treatment coincided with a decreased concentration of purine nucleotides (*Figure 4B*), while in EMB-treated cells we could measure very low levels of all dNTPs (*Figure 4C and H*). In the combination treatment, we could not measure significant differences (*Figure 4A*). The dGTP pool decreased in both absolute and relative terms across all treatments where dNTP pool changes were observed (*Figure 4B–F and G*, respectively). A smaller cell size coincides with a lower cellular dNTP concentration, while no clear correlation is observed between drug-induced cell length increase and dNTP pool expansion (*Figure 4I*).

## Stress-induced drug tolerance is developed upon pretreatment with the sublethal concentration of CIP

To compare the result of the mutation accumulation experiment to a phenotype-based drug resistance assay, we chose the fluctuation assay generally used in the literature (*Krašovec et al., 2019*). Mutation rates in these tests are calculated based on the difference in the number of CFU values between cultures grown in regular broth compared to those in selecting broths. These assays assume that the resistance exclusively occurs upon one mutation event. Since the genetic background of a drug-tolerant colony is not confirmed, this presumption potentially leads to a significant misinterpretation of the actual mutation rate. For clarity, we refer to the mutation rate estimations in our phenotype-based resistance assay as the tolerance rate. For a valid comparison with the results of our mutation accumulation assay, we installed similar experimental conditions. Specifically, culturing was done on agar plates, the applied drug concentrations were in the same range as used during the mutation accumulation process, then colonies were washed off and CFU counting plates were streaked from the resuspended bacteria (*Figure 5A*). We found that the estimated rate of emergence of the tolerance for CIP is three orders of magnitude higher than the mutation rate calculated based on WGS ($10^{-7}$ vs. $10^{-10}$, *Figure 2B*). Furthermore, following a 24–96 hr exposure to a sublethal 0.3 µg/ml dose of CIP, a phenotypic tolerance appears in a significant portion of the cells to an otherwise lethal 0.5 µg/ml dose (*Figure 5B*). The tolerant cell population increased with the length of the preincubation time before reaching a maximum (*Figure 5B*).

To confirm that the rapid increase in drug tolerance following short-term exposure to CIP is linked to non-genetic factors, we repeated the experiment using the 96 hr preincubation time for DNA isolation and WGS. After pretreatment, DNA was isolated from colonies on five parallel plates for each of the three biological replicates, followed by WGS (*Figure 5A*, *Figure 5—figure supplement 1*). In all measured samples, we detected a single mutation in a gene encoding an uncharacterized protein probably involved in lipid metabolism (MSMEG_6151; *Table 2*).

We also sequenced the genomes of colonies grown at the higher CIP concentration (0.5 µg/ml) and detected no mutations.

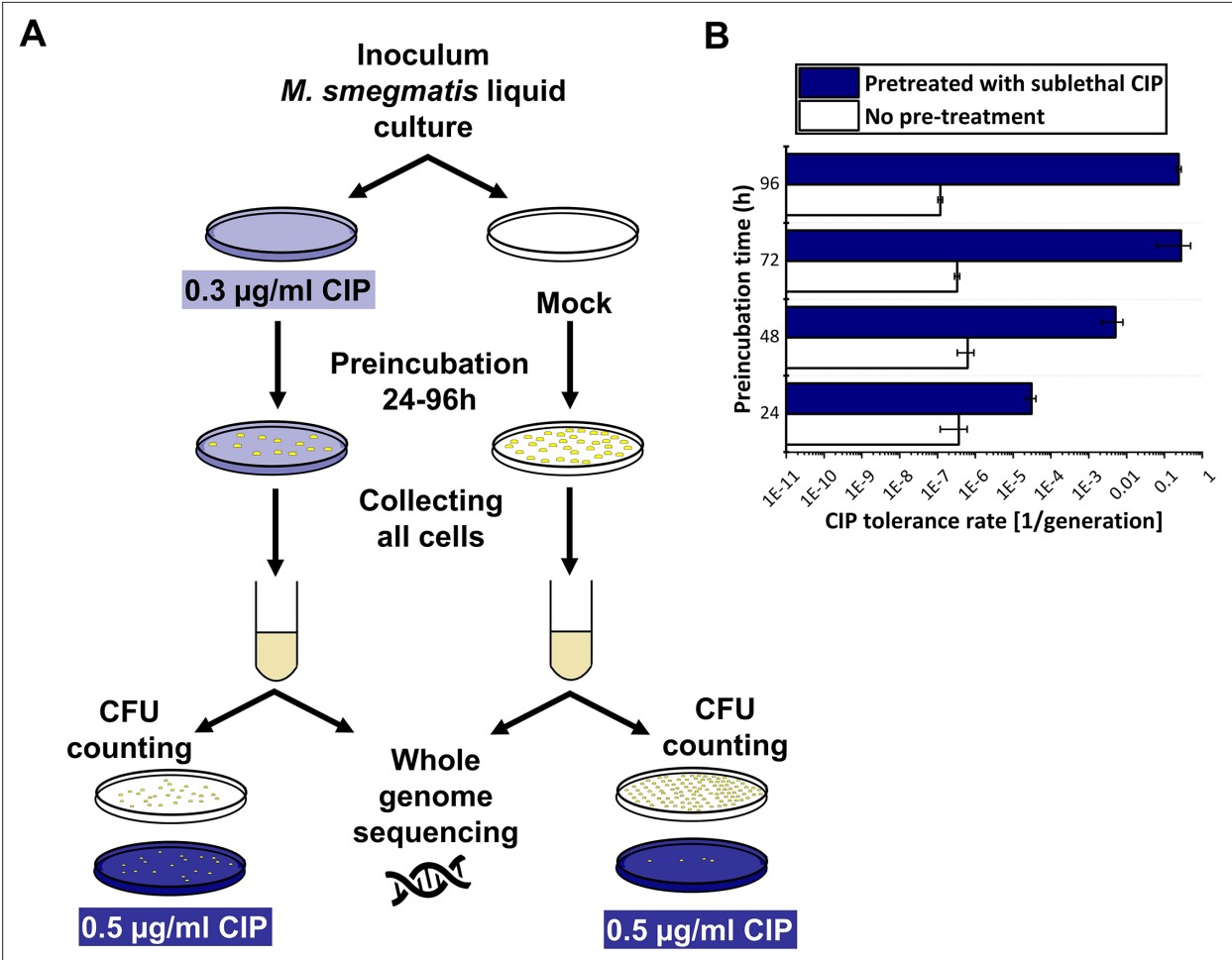

**Figure 5.** Phenotypic ciprofloxacin (CIP) tolerance assay. (**A**) Scheme of the fluctuation test used in the study. (**B**) Development of phenotypic resistance to a selecting CIP concentration following preincubation with a sublethal CIP concentration for various time periods. Data bars represent the averages of three biological replicates each carried out in three technical replicates; error bars represent SE.

The online version of this article includes the following figure supplement(s) for figure 5:

**Figure supplement 1.** Ciprofloxacin (CIP) tolerance of *M. smegmatis* preincubated for 96 hr on CIP-containing plates sent for Whole Genome Sequencing (WGS).

## Discussion

One of the reasons TB is still a great medical challenge is the frequent incidence of resistant cases. The main goal of our research was to get a better understanding of the molecular mechanisms of drug resistance development in mycobacteria. We started with the hypothesis that long-term exposure to first-line antitubercular drugs increases mutability.

### Drug resistance in *M. smegmatis* does not arise from increased mutation rates under antibiotic pressure

Measured and estimated mycobacterial mutation rates in the earlier literature are in the order of $10^{-10}$ /bp/generation (*Ragheb et al., 2013*, *Kucukyildirim et al., 2016*, *Ford et al., 2011*, *Colangeli et al., 2014*). This low constitutive mutation rate by itself does not explain the biological diversity observed in clinical isolates (*Sun et al., 2012*). This diversity might result from an elevated mutagenesis rate or the accumulation of different strains from the environment. We conducted a modelling study in *M. smegmatis* to investigate whether exposure to first-line antibiotics generates such biological diversity and if yes, by what possible molecular mechanism. We measured the appearance of drug-induced mutations in the genome in a mutation accumulation assay using WGS. We also examined the rapid

occurrence of phenotypic tolerance. The difference between the results of the phenotypic and the mutation accumulation studies was surprisingly large. Even without pretreatment, a tolerance rate on the order of $10^{-7}$/generation was observed for CIP, consistent with literature data from fluctuation assays (*Bergval et al., 2012*; *David, 1970*). However, in the mutation accumulation assay, the number of mutations did not change significantly compared to the untreated control. The mutation rate increase was only significant in the case of the UV treatment serving as a positive control for the experiments (*Figure 2B*). Previous studies claiming mutation rate increase upon antibiotics treatment assessed mutation rates using fluctuation assays and no direct evidence of the change in the genetic material was shown (*Gillespie et al., 2005*; *Kohanski et al., 2010*). However, it should be noted that David's study, which automatically classified bacteria growing in fluctuation assays as mutants without confirming genetic changes, also suggested that the term 'acquired resistance' in tubercle bacilli has only practical meaning and lacks experimental foundation (*David, 1970*). Our findings imply that the emergence of drug resistance in this study is solely attributed to phenotypic factors. Phenotypic changes upon antibiotic treatment have widely been investigated (*Briffotaux et al., 2019*) including potential bistability (*Dubnau and Losick, 2006*) and/or the upregulation of efflux pumps (*Calgin et al., 2013*; *Machado et al., 2012*). It is noteworthy that spontaneous mutagenesis is easily induced through UV treatment. Considering that mycobacterial species spread through air droplets, it is conceivable that the exposure of these droplets to environmental UV radiation could potentially lead to the generation of new mutations.

## The combination treatment with frontline drugs induces an overall upregulation in the DNA repair pathways aimed at eliminating misincorporations

The intracellular lifestyle of the TB pathogen implies that these bacteria must face various stress conditions and damaging agents including reactive oxygen and nitrogen species inside macrophages. Therefore, stress-induced transcriptional changes in mycobacteria have been studied on genome-wide scales (*Briffotaux et al., 2019*, *Li et al., 2017*) and one study found a specific activation of the DNA repair system in response to CIP similar to ours (*O'Sullivan et al., 2008*). Although *M. smegmatis* is not an intracellular pathogen, it shares the DNA repair pathways with *M. tuberculosis* and is often used to study how mycobacteria deal with DNA lesions (*Singh, 2017*). We focused our investigation on stress-induced transcriptional changes that may account for the protection of genomic integrity under the drug pressure of first-line antituberculotic drugs.

Redox potential change is a well-known and common phenotypic response to INH in mycobacteria (*Niki et al., 2012*). The downregulation of KatG1 and Nei2 in response to our INH treatment (*Figure 3*) is in line with this and might indicate a reduced cellular redox potential. KatG1 is the enzyme that activates the prodrug INH (*Niki et al., 2012*), therefore, the downregulation of this enzyme decreases the active drug concentration and increases the tolerance of *M. smegmatis* against INH. In the case of the first-line combination treatment, however, KatG1 was highly upregulated, indicating high ROS levels in the cell (*Wayne and Diaz, 1986*). High ROS levels are known to cause damage to nucleobases and the nucleotide pool is a major effector of oxidative stress-induced genotoxic damage (*Rai, 2010*). In line with this, we observed upregulation in dNTP pool sanitation, base- and nucleotide-repair pathways which play crucial roles in preventing and repairing DNA damage caused by oxidative stress. The observed synergistic effect clearly results from the combination of first-line drugs, as we did not observe this effect when applying the drugs individually. The observed upregulation of the relevant DNA repair enzymes might account for the low mutation rate even under drug pressure. Notably, error-prone polymerases DinB2 and DnaE2 exhibited significant upregulation without inducing a mutator phenotype. This indicates that error-prone and error-free repair mechanisms are coactivated, predominantly resulting in error-free repairs.

## dNTP pool alterations induced by frontline drugs neutralize each other in the combination treatment resulting in normal DNA precursor pools

The building blocks of DNA constitute a critical component within the molecular aspects of mutability. It has been shown that increased or imbalanced dNTP pools induce mutagenesis in prokaryotes *Gon et al., 2011* and eukaryotes (*Pai and Kearsey, 2017*). To assess the impact of drug treatment on dNTP pools and its correlation with genome stability, we quantified the concentrations of dNTPs

in cell extracts obtained from the drug-treated cells. When treating the cells with frontline drugs EMB and INH individually, the observed reductions in dNTP pool sizes and cell size (as illustrated in *Figure 4H–I*) aligned well with the concurrent downregulated transcript levels (*Figure 3*). Resting states of bacteria have also been characterized by a decrease in cell size and dATP levels (*Rittershaus et al., 2013*; *Wu et al., 2016*). These observations thus probably reflect the bacteriostatic effect of these drugs causing metabolic processes to enter a dormant state, accompanied by the downregulation of enzymes involved in dNTP synthesis. The combined treatment yielded the least significant alteration from the untreated control compared to all monotreatments (*Figure 4*). An elevation in the dNTP pools during cytostatic or cytotoxic treatment is more unexpected and suggests elevated DNA repair activity. This observation, particularly in the case of CIP treatment, aligns with the substantial increase in the expression of DNA repair synthesis genes, as depicted in *Figure 3*. Among all administered treatments, only the CIP treatment led to a notable dNTP imbalance and a substantial overall rise in dNTP pools, due to elevated levels of dTTP and dATP. This coincides with the largest changes in the expression of DNA repair genes, particularly those associated with the SOS response and homologous recombination (*Figure 3*). Interestingly, the dGTP level decreased with all drug treatments. This finding suggests that dGTP may play a role in a general stress response. It is noteworthy that not all dNTP imbalances are created equal. Specifically, an excess of dGTP has been identified as a significant contributor to mutations (*Martomo and Mathews, 2002*; *Schmidt et al., 2019*). It must be noted that in these (and most) organisms dGTP is the least abundant among dNTPs. However, in mycobacteria, a unique scenario exists where dGTP is the most abundant dNTP species (*Pancsa et al., 2022*) and mycobacterial genomes are characterized by a high GC content (*Andersson and Sharp, 1996*). A reduction in dGTP levels in this context may contribute to minimizing DNA lesions by enhancing proofreading efficiency.

## Our results do not support drug resistance acquisition through drug-induced microevolution

Our hypothesis that systematic antibiotics treatment induces mutation rate increase in *M. smegmatis* failed, as we did not observe any significant impact of antibiotics on mutability in laboratory conditions. Only in the case of CIP treatment, a second-line TB drug known for directly inducing DNA damage, could we detect a slightly (but not significantly) elevated mutation rate. The treatment of *M. smegmatis* with the clinically used combination therapy drugs did not induce a mutator effect, quite the opposite. The observed activation of DNA repair processes likely mitigates mutation pressure, ensuring genome stability. However, to confirm this hypothesis, these investigations should be conducted using genetically modified DNA repair mutant strains.

If there is no indication for a priori drug resistance, TB patients are treated with the combination therapy of first-line antituberculotics. In at least 17% of the treatments, resistance to RIF or RIF+INH (called multidrug resistance) emerges (*World Health Organization, 2023*). There are two models for the development of drug-resistant TB: acquired and transmitted drug resistance. The acquired drug resistance model suggests that resistance is developed within patients with active TB through microevolution (*Ley et al., 2019*). Several studies suggest examples of microevolution (*Al-Hajoj et al., 2010*, *Ssengooba et al., 2016*) especially those involving the hypermutable *Beijing Mtb* lineage (*Hakamata et al., 2020*). However, it is crucial to note that distinguishing between acquired and transmitted resistance is not straightforward based solely on allele variants found in the sputum. In the transmitted resistance model, a patient accumulates a pool of mycobacteria with different genotypes during latent infection. This population mix is essentially clonal, as *M. tuberculosis* strains possess a highly conserved core genome (*Gray and Derbyshire, 2018*), but with several genetic allele variants having limited representation. The transition of the disease to an active phase, along with subsequent chemotherapy, leads to adaptive selection from the pre-existing pool of variants. The concept that certain TB cases involve mixed infections has been substantiated in clinical cases using phage typing and whole-genome sequencing (*Bates et al., 1976*; *Boritsch and Brosch, 2017*). The transmissibility of resistant variants has been confirmed through strain-specific PCR (*Braden et al., 2001*), and selective adaptation in a patient during chemotherapy has also been demonstrated (*Hingley-Wilson et al., 2013*). Furthermore, it has been shown that clonal complexity is reduced by culturing, leading to the underrecognition of polyclonal infections in culture-based diagnosis (*Martín et al., 2010*). The WHO estimates that a quarter of the world's population is latently infected by *M. tuberculosis*, accumulating

different TB strains throughout their lives (*World Health Organization, 2021*). Consequently, patients may harbor high heterogeneity, facilitating the spread and fixation of a genetic variant with some advantage in specific environmental conditions.

We acknowledge the limitations of using *M. smegmatis* as a model for the intracellular pathogen *M. tuberculosis*, which is associated with complex pathology. Nevertheless, given the conserved molecular mechanisms of genome maintenance in mycobacteria, we can conclude that the mycobacterial genome is not prone to microevolution upon prolonged exposure to the antibiotics employed in our study and the clinics.

# Materials and methods

**Key resources table**

| Reagent type (species) or resource | Designation | Source or reference | Identifiers | Additional information |
|---|---|---|---|---|
| Strain, strain background (*Mycobacterium smegmatis*) | mc2-155 | *Snapper et al., 1990* | GenBank: NC_008596.1 | |
| Other | DAPI stain | Sigma | D9542 | 10 µg/ml |
| Chemical compound, drug | Isoniazid | Sigma | I3377 | |
| Chemical compound, drug | Ethambutol | Sigma | E4630 | |
| Chemical compound, drug | Rifampicin | Sigma | R3501 | |
| Chemical compound, drug | Pyrazinamide | Sigma | 40751 | |
| Chemical compound, drug | Ciprofloxacin | Sigma | 17850 | |
| Chemical compound, drug | Mytomicin-C | Sigma | 10107409001 | |
| Commercial assay or kit | phenol:chloroform:IAA (25:24:1) | Sigma | Sigma: 3803 | For genomic DNA extraction |
| Commercial assay or kit | Whole genome sequencing | Novogene Ltd., Beijing, China | | Executed on Illumina 1.9 instruments with 600-basepair fragments as 2×150 bp paired-end sequencing |
| Commercial assay or kit | RNeasy Mini kit | Qiagen | Qiagen: 74524 | Used with RNA protect bacteia reagent (Qiagen: 76506) and DNAse I (Qiagen: 79254) |
| Commercial assay or kit | High-Capacity cDNA Reverse Transcription Kit | Applied Biosystems | Applied Biosystems: 4374967 | 95–105 ng total RNA was used for each reaction |
| Other | Mytaq PCR premix | Bioline | Bioline: 25046 | For qPCR measurements |
| Other | EvaGreen | VWR | VWR: #31000 | For qPCR measurements |
| Software, algorithm | NucleoTIDY | *Szabó et al., 2020*; http://nucleotidy.enzim.ttk.mta.hu | V1.8 | |
| Other | TEMPase Hot Start DNA Polymerase | VWR | VWR: 733–1838 | For dNTP measurements |
| Other | methanol | Sigma | | For dNTP isolation |
| Sequence-based reagent | NDP-1 | *Szabó et al., 2020* | Primer for dNTP measurement | CCGCCTCCACCGCC |
| Sequence-based reagent | FAM-dTTP | *Szabó et al., 2020* | Probe for dTTP measurement | 6-FAM/ AGGACCGAG/ZEN/GCAAGAGCGAGCGA / IBFQ |

*Continued on next page*

*Continued*

| Reagent type (species) or resource | Designation | Source or reference | Identifiers | Additional information |
|---|---|---|---|---|
| Sequence-based reagent | FAM-dATP | *Szabó et al., 2020* | Probe for dTATP measurement | 6-FAM/ TGGTCCGTG/ZEN/GCTTGTGCGTGCGT /IBFQ |
| Sequence-based reagent | FAM-dGTP | *Szabó et al., 2020* | Probe for dTGTP measurement | 6-FAM/ ACCATTCAC/ZEN/CTCACACTCACTCC /IBFQ |
| Sequence-based reagent | FAM-dCTP | *Szabó et al., 2020* | Probe for dTCTP measurement | 6-FAM/ AGGATTGAG/ZEN/GTAAGAGTGAGTGG /IBFQ |
| Sequence-based reagent | dTTP-DT1 | *Szabó et al., 2020* | Template oligo for dTTP measurement | TCGCTCGCTCTTGCCTCGGTC CTTTATTTGGCGGTGGAGGCGG |
| Sequence-based reagent | dATP-DT1 | *Szabó et al., 2020* | Template oligo for dATP measurement | ACGCACGCACAAGCCACGGAC CAAATAAAGGCGGTGGAGGCGG |
| Sequence-based reagent | dCTP-DT1 template | *Szabó et al., 2020* | Template oligo for dCTP measurement | CCACTCACTCTTACCTCAATCCTTT GTTTGGCGGTGGAGGCGG |
| Sequence-based reagent | dGTP-DT2 template | *Szabó et al., 2020* | Template oligo for dATP measurement | GGAGTGAGTGTGAGGTGAATGGTT TCTTTCTTTGGCGGTGGAGGCGG |
| Software | FastQC | Babraham Bioinformatics https://www.bioinformatics.babraham.ac.uk/projects/fastqc/ | | v.0.11.9 |
| Software | Trimmomatic | *Bolger et al., 2014*; http://www.usadellab.org/cms/?page=trimmomatic | | Trimmomatic-0.38 |
| Software | Bowtie2 | *Langmead and Salzberg, 2012*; https://bowtie-bio.sourceforge.net/bowtie2/index.shtml | | 2.5.4 |
| Software | Samblaster | *Faust and Hall, 2014*; https://github.com/GregoryFaust/samblaster | | 0.1.26 RRID:SCR_000468 |
| Software | Samtools | *Li et al., 2009*; https://www.htslib.org/ | | 1.20 |
| Software | Picard | https://github.com/broadinstitute/picard | | 2.23.3 RRID:SCR_006525 |
| Software | GATK | *McKenna et al., 2010*; https://gatk.broadinstitute.org/hc/en-us | | 4.1.8.1 |

## Bacterial strains, media, and growth conditions

*M. smegmatis* mc$^2$155 (*Snapper et al., 1990*) strains were grown in Lemco broth (5 g/l Lab-Lemco, 5 g/l NaCl, 10 g/l Bacto peptone, 0.05% Tween-80) or on solid Lemco plates (6.25 g/l Lab-Lemco,

6.25 g/l NaCl, 12.5 g/l Bacto peptone, 18.75 g/l Bacto agar).

## Optimization of stress treatment conditions in liquid cultures and agar plates

The applied concentrations of drugs were optimized using serial dilutions of the compounds. In the case of liquid cultures, we monitored growth on a logarithmic scale by measuring the number of colony-forming units (CFU) or the optical density (OD) at 600 nm (*Figure 1—figure supplement 1*). The PZA treatments were done in acidic broth (pH = 5.5 set using HCl). For agar plates, we determined the CFU of untreated mid-exponential phase (OD = 0.4–0.5) liquid cultures on both non-selective and drug-containing agar plates (*Figure 1—figure supplement 2*). We also monitored cell morphology in response to drug treatment. For further experiments, sublethal concentrations of drugs were chosen to obtain an adequate quantity of research material (DNA, RNA, dNTP) for downstream analysis while the effect of the treatment was clearly indicated by a decrease of viability and/or change in cell size and morphology. The concentrations of applied drugs and stress conditions are compiled in *Table 1*.

## Stress treatment in liquid cultures

Cells were grown in 100 ml liquid culture until an OD (600)=0.1±0.02 was reached, then the appropriate quantity of drug (*Table 1*) was added to half of the cultures. The other half of the same culture was used as a control. We conducted the treatments for 8 hr. The cultures were then centrifuged (20 min, 3220 g, 4 °C) and the resulting pellets were used for downstream analysis. The total CFUs were determined for each culture. The generation time after the treatments was calculated using the formula:

$$T_d = t / \log_2 (N_t / N_0),$$

where $T_d$ is the generation time, t is the time interval between measurements, and $N_t$ and $N_0$ are the final and initial population sizes, respectively.

## Microscopic analysis of cell morphology upon treatments

For morphological studies, 200–200 µl stress-treated and control cells were retrieved before RNA or dNTP extraction and washed with PBS containing 0.1% Triton X-100. The cells were then fixed in 4% PFA dissolved in PBS for 30 min at 37 °C. Cells were stained with 10 µg/ml DAPI for 30 min at 37 °C, then streaked onto microscopy slides covered with 0.1% low melting agarose (Sigma). Imaging was done using phase-contrast and fluorescent modes on a Leica DM IL LED (Leica) microscope. The cell size and volume were quantified using the automated recognition of the BacStalk software (https://drescherlab.org/data/bacstalk/; *Hartmann et al., 2020*). The cell length distribution diagram was prepared using OriginPro 2018 (OriginLab Corporation, Northampton, MA, USA.). The sample size, calculated means, and standard deviations are compiled in *Figure 1—source data 1*.

## Mutation accumulation (MA) experiments

Sixteen independent *M. smegmatis* mc[2] 155 MA lines were initiated from a single colony for every treatment. The ancestor cell colony was generated by streaking a new single colony from plate to plate five times before the beginning of the treatments to ensure a single common ancestor. Lemco agar medium was used for the MA line transfers. The specific stress treatment conditions are summarized in *Table 1*. All MA lines were incubated at 37 °C. Every 3 days, a single isolated colony from each MA line was transferred by streaking to a new plate, ensuring that each line regularly passed through a single-cell bottleneck (*Kibota and Lynch, 1996*). Treatments were performed for 60 days. We calculated 6.3±0.35 hr of generation time on the plate in this experimental setup. Thus, each line passed through ~230 cell divisions. Some mock treatments were performed for 120 days to ensure a presumably sufficient number of mutational events without stress treatment. Following the MA procedure, a single colony was transferred from all strains to a new plate without stress treatment and grew for another 3 days for expansion. Frozen stocks of all lineages were prepared in 20% glycerol at −80 °C.

## Assessment of drug tolerance following MA experiments

The development of tolerance to the applied treatment was assessed by measuring the minimal inhibitory concentration (MIC) of both the mock-treated and stressed MA strains. Three randomly chosen strains from both the mock-treated and stress-treated groups were resuscitated on plates containing the same stress conditions as those used in the MA experiment. Liquid cultures were inoculated and diluted to an OD(600) of 0.001 in sterile, round-bottom 96-well plates (Sarstedt). The wells contained the specific drug in serial dilution for both the stressed strains and control samples. Cells were grown at 37 °C without agitation. Plates were scanned and analyzed, and MIC values were determined based on the last well in which cell growth was observed.

## DNA extraction

A single colony was inoculated into 10 ml liquid culture from all lineages, was grown until $OD_{600} = 0.8–1.0$, and harvested. For genomic DNA purification, five or six grown cultures of individual lineages from the same treatment with identical estimated cell numbers (based on OD measurements) were pooled before isolation. For cell disruption, the cells were resuspended in 1 ml of 10 mM Tris, pH 7.5, and 0.1 mm glass beads were added to a final volume of 1.5 ml. The cells were disrupted using a cell disruptor (Scientific Industries SI-DD38 Digital Disruptor Genie Cell Disruptor) in a cold room (at 4 °C). After centrifugation for 10 min at 3220 g, and at room temperature, DNA was extracted from the supernatant by phenol:chloroform:IAA (25:24:1) extraction followed by isopropanol precipitation. The quality and quantity of the extracted DNA was evaluated using UV photometry in a Nanodrop-2000 instrument and by agarose gel electrophoresis.

## DNA library preparation and whole genome sequencing

The DNA library preparation and whole genome sequencing (WGS) was done at Novogene Ltd., Beijing, China. Sequencing was executed on Illumina 1.9 instruments with 600-basepair (bp) fragments as 2×150 bp paired-end sequencing. An average read depth of 267 was achieved across all samples.

## WGS analysis and mutation identification

Three parallel pooled samples were sequenced for every treatment, each contained five or six individually treated MA lineages that add up to a subtotal of 15–18 individual lineages. FastQC was used to analyse the quality of the raw reads. In case if adapters and low-quality bases (Phred score <20) were present in the samples, bases were trimmed with Trimmomatic (*Bolger et al., 2014*). We mapped our paired-end reads to *M. smegmatis* mc$^2$ 155 reference genome (GenBank accession number: NC_008596.1) by Bowtie2 (*Langmead and Salzberg, 2012*). PCR duplicates were removed with the use of Samblaster (*Faust and Hall, 2014*). We converted SAM files to BAM files, and sorted them with SAM tools (*Li et al., 2009*). Read groups were replaced by the Picard tool. Single nucleotide variations (SNVs), insertions, and deletions were called from each alignment file using the Haplotype-Caller function of the Genome Analysis Toolkit (*McKenna et al., 2010*). We analyzed the frequency of occurrence (% of all reads of a pooled sample) of each SNV, insertions, and deletions (hits) with our in-house Python scripts and compared it to the frequency of occurrence of the same hits in every other lineage. We considered mutations as spontaneously generated mutations only in case if no other lineages carried that variant in any depth and if hits reached at least 6% frequency of the reads at the corresponding position (theoretically, a spontaneously generated mutation in a pooled sample emerges with 20% or 16.7% frequency when five or six lineages are pooled, respectively, however, we allowed some variety when choosing 6% as a lower limit and 39.9% as an upper limit). Sequencing data are available at European Nucleotide Archive (ENA) with PRJEB71590 project number. Please note that we incorporated some of our additional sequencing data into the analysis, curated under the umbrella project at the ENA along with the present dataset.

## RNA isolation and cDNA synthesis

For RNA extraction, cell pellets were resuspended in 2 ml RNA protect bacteria reagent (Qiagen; cat. no.:76506), incubated for 5 min at room temperature, and centrifuged for 20 min at 3220 g and at 4 °C before storage at –80 °C. Total RNA extraction was performed with the Qiagen RNeasy Mini kit (cat. no.: 74524). To disrupt cells, 5×1 min of vortexing with glass beads in the manufacturer's lysis

buffer was performed followed by 1 min poses on ice. DNase digestion was performed on a column with Qiagen DNase I (cat. no.: 79254), for 90 min at room temperature. For quantitative and qualitative RNA analysis, spectrometry by Nanodrop 2000 and non-denaturing 1% agarose gel electrophoresis (50 min/100 V) were performed, respectively. cDNA synthesis was performed using the Applied Biosystems High-Capacity cDNA Reverse Transcription Kit with RNase Inhibitor (cat. no.: 4374967). 95–105 ng total RNA was used for each reaction.

## Choosing the reference genes for the study

We tested SigA (MSMEG_2758), Ffh (MSMEG_2430), and ProC (MSMEG_0943) as possible reference gene candidates. SigA is a widely used reference gene in prokaryotes (*Hirmondo et al., 2017*; *Madikonda et al., 2020*; *Milano et al., 2004*) Ffh and ProC genes are shown to be stably expressed in other pathogens (*Gomes et al., 2018*). Using GeNorm (*Fu et al., 2020*; *Sundaram et al., 2019*) analysis, SigA and Ffh proved to be stably expressed in our experimental system (*Figure 3—figure supplement 1*).

## Gene expression quantification

qPCR measurements were performed on a Bio-Rad CFX96 Touch Real-Time PCR Detection System. Primers were designed using IDT DNA oligo customizer (https://eu.idtdna.com/), and were produced by Sigma Aldrich (for sequences, see *Supplementary file 1*). The qPCR reaction mixtures contained 7–7 nmoles of forward and reverse primers, 0.25 µl of the cDNA, Bioline Mytaq PCR premix (cat. no.: 25046), and VWR EvaGreen (cat. no.: #31000) in a total reaction volume of 10 µl. The thermal profile was as follows: 95 °C/10 min, 50 x (95 °C/10 s; 62 °C/10 s; 72 °C/10 s). Melting curves were registered between 55 °C and 95 °C with an increment of 0.5 °C (*Figure 3—figure supplement 2*). The applied primers and their measured efficiencies are compiled in *Supplementary file 1*. The qPCR data were analyzed using the Bio-Rad CFX Maestro software and numerically shown in *Figure 3—source data 1*. Non-reverse transcribed controls and no-template controls were used to account for any irrelevant DNA contamination. three technical, and three biological replicates were used for all measurements.

## dNTP extraction

dNTP extraction and measurement were performed according to (*Szabó et al., 2020*). Briefly, the cell pellets were extracted in precooled 0.5 ml 60% methanol overnight at −20 °C. After 5 min of boiling at 95 °C, the cell debris was removed by centrifugation (20 min, 13,400 g, 4 °C). The methanolic supernatant containing the soluble dNTP fraction was vacuum-dried (Eppendorf) at 45 °C. Extracted dNTPs were dissolved in 50 µl nuclease-free water and stored at −20 °C until use.

## Determination of the cellular dNTP pool size

Determination of the dNTP pool size in each extract was as follows: 10 pmol template oligo (Sigma), 10 pmol probe (IDT), and 10 pmol NDP1 primer (Sigma) (see sequences in key resources table and *Table 3*) was present per 25 µl reaction. The concentration of each non-specific dNTP was kept at

**Table 3.** Oligonucleotides used for the dNTP measurements.

| Name | Sequence (5'→3') |
|------|------------------|
| NDP-1 primer | CCGCCTCCACCGCC |
| FAM-dTTP probe | **6-FAM**/AGGACCGAG/**ZEN**/GCAAGAGCGAGCGA/**IBFQ** |
| FAM-dATP probe | **6-FAM**/TGGTCCGTG/**ZEN**/GCTTGTGCGTGCGT/**IBFQ** |
| FAM-dGTP probe | **6-FAM**/ACCATTCAC/**ZEN**/CTCACACTCACTCC/**IBFQ** |
| FAM-dCTP probe | **6-FAM**/AGGATTGAG/**ZEN**/GTAAGAGTGAGTGG/**IBFQ** |
| dTTP-DT1 template | TCGCTCGCTCTTGCCTCGGTCCTTT**A**TTTGGCGGTGGAGGCGG |
| dATP-DT1 template | ACGCACGCACAAGCCACGGACCAAA**T**AAAGGCGGTGGAGGCGG |
| dCTP-DT1 template | CCACTCACTCTTACCTCAATCCTTT**G**TTTGGCGGTGGAGGCGG |
| dGTP-DT2 template | GGAGTGAGTGTGAGGTGAATGGTTT**C**TTT**C**TTTGGCGGTGGAGGCGG |

100 µM. VWR TEMPase Hot Start DNA Polymerase (VWR) was used at 0.9 unit/reaction in the presence of 2.5 mM MgCl$_2$. To record calibration curves, the reaction was supplied with 0–12 pmol specific dNTP. Fluorescence was recorded at every 13 s in a Bio-Rad CFX96 Touch Real-Time PCR Detection System or in a QuantStudio 1 qPCR instrument. The thermal profile was as follows: 95 °C 15 min, (60 °C 13 s)×260 cycle for dATP measurement, and 95 °C 15 min, (55 °C 13 s)×260 cycle for dTTP, dCTP, and dGTP measurements. Results were analyzed using the nucleoTIDY software (http://nucleotidy.enzim.ttk.mta.hu/; *Szabó et al., 2020*;). Results were given in molar concentrations for better comparison. To this end, cell volumes were calculated using the BacStalk software based on microscopic images for every treatment. Besides the graphical presentation of the result, numerical data can be found in *Figure 4—source data 1*.

### Tolerance assay

We used a modified version of fluctuation assays (*Krašovec et al., 2019*) for the estimation of the rate of emergence of tolerant cells upon preincubation with a sublethal dose of CIP (0.3 µg/ml). An initial 100 ml culture was grown to OD =0.4–0.5 (three biological replicates), was centrifuged for 30 min at 800 g and at 4 °C, then resuspended in 5 ml Lemco. 100 µL from this stock solution was streaked and cultured on a normal Bacto Agar plate, and Bacto Agar containing 0.3 µg/ml CIP. Parallel plates were incubated for 4, 24, 48, 72, and 96 hr at 37 °C. Colonies were washed off the plate with 6 ml Lemco broth by incubation for 30 min on a rocking shaker. Then CFU was determined on Bacto agar plates containing 0.5 µg/ml CIP, and non-selective Bacto agar plates. Tolerance rates were calculated using the following formula:

$$tolerance_t = \frac{\dfrac{CFU_{res.}^t - CFU_{res.}^0}{CFU_{total}^t - CFU_{total}^0}}{\dfrac{t}{log_2\left(\dfrac{CFU_{total;t}}{CFU_{total;0}}\right)}}\left[\frac{1}{generationtime\,(h)}\right]$$

$t$: time of preincubation
$CFU_{total}^0$: Number of colonies on non-selecting agar plates at reference time point
$CFU_{total}^t$: Number of colonies on non-selecting agar plates at t hours
$CFU_{res.}^0$: Number of colonies on plates containing 0.5 µg/ml CIP at reference time point
$CFU_{res.}^t$: Number of colonies on plates containing 0.5 µg/ml CIP at t hours
$CFU_{res.}^t - CFU_{res.}^0$: Number of newly emerging resistant colonies
$CFU_{total}^t - CFU_{total}^0$: Number of growing colonies
$\dfrac{t}{log_2\left(\frac{CFU_{total;t}}{CFU_{total;0}}\right)}$: Generation time during pre-treatment

The tolerance assay was repeated on 15 parallel plates for each biological replicate to obtain enough cells for genomic DNA extraction for WGS. Plates containing 0.3 µg/ml CIP were incubated for 96 hr at 37 °C. Colonies were washed off the plates with 6 ml of Lemco broth. Genomic DNA was isolated and sent to WGS. CFU was also determined on Bacto Agar plates containing 0.5 µg/ml CIP and non-selective Bacto Agar plates (*Figure 5—figure supplement 1*).

### Statistics

We used an initial F-test to test the equality of variances of the tested groups. If the F-test hypothesis was accepted (p<0.05), we used the two-way homoscedastic t-probe; if rejected, we used the two-way Welch's t-probe to assess differences at a significance level p<0.05 if not stated otherwise. F- and t-statistics were counted for the ΔCt values (*Yuan et al., 2006*) of the qPCR results and for the concentrations normalized to the cell volume in the case of the dNTP measurements. For the statistical analysis of the mutation rates, we used the t-test on the natural logarithm of the obtained mutation rate values.

## Acknowledgements

We would like to express our gratitude for the assistance provided by Ádám Póti in the analysis of the WGS data. Funding this work was supported by the National Office for Research and Technology,

Hungary [grant number OTKA-K115993 and OTKA-K138318 to JT, OTKA-PD128254 to RH]. Funding for open access charge: National Office for Research and Technology, Hungary.

## Additional information

### Funding

| Funder | Grant reference number | Author |
|---|---|---|
| National Office for Research and Technology, Hungary | K115993 | Judit Toth |
| National Office for Research and Technology, Hungary | PD128254 | Rita Hirmondó |
| National Office for Research and Technology, Hungary | K138318 | Judit Toth |

The funders had no role in study design, data collection and interpretation, or the decision to submit the work for publication.

### Author contributions

Dániel Molnár, Conceptualization, Investigation, Visualization, Writing – original draft, Writing – review and editing; Éva Viola Surányi, Conceptualization, Data curation, Software, Formal analysis, Investigation, Visualization, Writing – original draft, Writing – review and editing; Tamás Trombitás, Dóra Füzesi, Formal analysis, Investigation; Rita Hirmondó, Conceptualization, Formal analysis, Supervision, Investigation, Visualization, Writing – original draft, Writing – review and editing; Judit Toth, Conceptualization, Formal analysis, Supervision, Funding acquisition, Investigation, Visualization, Writing – original draft, Writing – review and editing

### Author ORCIDs

Judit Toth ⓘ https://orcid.org/0000-0002-0965-046X

Reviewer #1 (Public review): https://doi.org/10.7554/eLife.96695.3.sa1
Reviewer #2 (Public review): https://doi.org/10.7554/eLife.96695.3.sa2
Reviewer #3 (Public review): https://doi.org/10.7554/eLife.96695.3.sa3
Author response https://doi.org/10.7554/eLife.96695.3.sa4

## Additional files

### Supplementary files

• Supplementary file 1. Nucleotide sequence and measured efficiency of primers used for the qPCR.

• Supplementary file 2. User guide for whole genome sequencing (WGS) data files deposited in the European nucleotide archive (ENA).

• MDAR checklist

### Data availability

The data underlying this article are available in the Figshare repository (DOI:10.6084/m9.figshare.26585884). Sequencing data are available at European Nucleotide Archive (ENA) with accession PRJEB71590.

The following datasets were generated:

| Author(s) | Year | Dataset title | Dataset URL | Database and Identifier |
|---|---|---|---|---|
| ÉV Surányi | 2024 | WGS on antibiotics-challenged Mycobacterium smegmatis | https://www.ebi.ac.uk/ena/browser/view/PRJEB71590 | Array Express, PRJEB71590 |
| Hirmondó R | 2024 | Supplementary datasets for the eLife manuscript "Genetic Stability of Mycobacterium smegmatis under the Stress of First-Line Antitubercular Agents: Assessing Mutagenic Potential" | https://doi.org/10.6084/m9.figshare.26585884 | figshare, 10.6084/m9.figshare.26585884 |

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
