## [Editor Report · eLife Assessment]

This **useful** study reports on the impact of antibiotic pressure on the genomic stability of the mc2155 strain of Mycobacterium smegmatis, a model for Mycobacterium tuberculosis. The findings of the study indicate that exposure to antibiotics did not lead to the development of new adaptive mutations in controlled laboratory environments, challenging the notion that antibiotic resistance arises from drug-induced microevolution. The genomic analysis provides detailed insights into the stability of M. smegmatis following exposure to standard TB treatment antibiotics, and the evidence suggesting that antibiotic pressure does not contribute to the emergence of new adaptive mutations is **solid**.

---

## [Referee Report · Reviewer #1 (Public review)]

Molnar, Suranyi and colleagues have generated a useful dataset characterizing the rate of mutations in Mycobacterium smegmatis - a non-pathogenic model mycobacterial strain, to several antibiotics at sub-lethal dose. The whole genome sequencing approach used is a strength of this study. Overall, the results are consistent with a low rate of mutations, consistent with other reports in Mycobacterium smegmatis and in vitro and clinical studies with Mycobacterium tuberculosis. The data supports phenotypic tolerance rather than genetic mutations as a driver.

The revised manuscript is improved and addresses several concerns raised by the reviewers from the previous rounds. These relate primarily to the presentation of data in the figures, but there is also new data in Figure 2 to show an increased MIC for M. smegmatis under antibiotic pressure. An additional dataset of sequences from ciprofloxacin-treated bacteria has also been generated and made publicly accessible, which will be of interest to the community.

---

## [Referee Report · Reviewer #2 (Public review)]

Summary

In this study, the authors evaluate the impact of selective pressure from chemotherapeutic drugs on the development of drug resistance in Mycobacteria, specifically through the acquisition of genetic mutations or phenotypic tolerance. Their findings indicate that treatment with sublethal concentrations of first-line antibiotics does not lead to enhanced mutation rates.

Strengths

The use of the mutation accumulation assay demonstrating low spontaneous mutation rates combined with the display of an increased MIC supports drug resistance as a consequence of phenotypic tolerance. Additionally, the use of the ciprofloxacin tolerance assay in combination with whole genome sequencing demonstrating a lack of mutations provides further support of this. The results now support the authors claims.

Weaknesses

Besides an increase in DNA stress response other underlying tolerance mechanisms were not established - increased efflux pump, thickening of the cell wall, decrease in metabolic processes, rerouting of metabolic processes etc.

---

## [Referee Report · Reviewer #3 (Public review)]

Summary:

This manuscript describes how antibiotics influence genetic stability and survival in Mycobacterium smegmatis. Prolonged treatment with first-line antibiotics did not significantly impact mutation rates. Instead, adaptation to these drugs appears to be mediated by upregulation of DNA repair enzymes. While this study offers robust data, findings remain correlative and fall short of providing mechanistic insights.

Strengths:

The strength of this study is the use of genome-wide approaches to address the specific question of whether or not mycobacteria induce mutagenic potential upon antibiotic exposure.

Comments on revised version:

The authors responded adequately to my comments, and I have no further suggestions for the revised manuscript.

---

## [Author Response]

The following is the authors’ response to the original reviews.

**Reviewer #1 (Public Review):**
In this manuscript, Molnar, Suranyi and colleagues have probed the genomic stability of Mycobacterium smegmatis in response to several anti-tuberculosis drugs as monotherapy and in combination. Unlike the study by Nyinoh and McFaddden http://dx.doi.org/10.1002/ddr.21497 (which should be cited), the authors use a sub-lethal dose of antibiotic. While this is motivated by sound technical considerations, the biological and therapeutic rationale could be further elaborated.

In the mutation accumulation experiments, we needed to ensure continuous and reproducible growth of a small number of colonies across multiple passages. This technical requirement necessitated the use of sublethal drug concentrations. However, sublethal doses also have biological relevance. Noncompliance with prescribed antibiotic regimens and the presence of antibiotic residues in food due to the extensive use of antibiotics in agricultural mass production are two obvious sources of prolonged exposure to sublethal antibiotics.

The results the authors obtain are in line with papers examining the genomic mutation rate in vitro and from patient samples in Mycobacterium tuberculosis, in vitro in Mycobacterium smegmatis and in vitro in Mycobacterium tuberculosis (although the study by HL David (PMID: 4991927) is not cited). The results are confirmatory of previous studies.

The two cited studies, along with several others, did not distinguish between genetic mutations and phenotypic responses to drug exposure (the fluctuation test alone is not suitable for this). Therefore, their objectives are not comparable to ours, which specifically investigated whether resistant colonies carry adaptive mutations. Nevertheless, we acknowledge the relevance of these studies and have now cited them in the appropriate sections in the text.

It is therefore puzzling why the authors propose the opposite hypothesis in the paper (i.e antibiotic exposure should increase mutation rates) merely to tear it down later. This straw-man style is entirely unnecessary.

The phenomenon of stress-inducible mutagenesis in bacterial evolution remains a topic of heated debate. The emergence of genetically encoded resistance may stem from either microevolution or the dissemination of pre-existing variants from polyclonal infections under drug pressure. We believe that the Introduction presents both of these hypotheses in a balanced manner to elucidate the rationale behind our mutation accumulation investigations.

The results on the nucleotide pools are interesting, but the statistically significant data is difficult to identify as presented, and therefore the new biological insights are unclear.

We now indicate statistical significance in the figure, in addition to the detailed statistical analysis of all dNTP measurements provided in Table S5.

Finally, the authors show that a fluctuation assay generates mutations with higher frequencies that the genetic stability assays, confirming the well-known effect of phenotypic antibiotic resistance.

What we show is that the fluctuation assay generated bacteria that tolerated the applied antibiotic without developing mutations. Conclusions about mutation rates are often drawn from fluctuation assays without confirming genetic-level changes, a discrepancy that persists despite these assays accounting for both phenotypic and genotypic alterations. By combining genome sequencing with fluctuation assays, our approach emphasizes the importance of distinguishing between these changes. While fluctuation assays remain valuable, inexpensive, and simple tools for evaluating the response of bacterial populations to various selective environments, they should not be considered definitive indicators of genetic changes.

**Recommendations For The Authors:**
The quality of the figures can be significantly improved. In Figure 1, cell lengths can be shown on separate histograms or better still as violin plots to enable better comparisons.

Thank you for the suggestion. We have revised the data presentation accordingly.

Details for statistical tests should be provided in the figure legend.

Statistical details are now added in the figure legend.

In Figure 2, the number of data points is not mentioned.

Statistical information is now added to the new Figure 2, which has been revised extensively based on suggestions from all Referees.

The data in Figure 3 would be much easier to comprehend as a heatmap.

The figure we provided is a color gradient table representing different gene expression levels, along with numerical data and statistical significance indicated within the color boxes, expanding the information content of a traditional heatmap. In response to the Referee's suggestion, we also prepared a hierarchical clustering heatmap, demonstrating that the grouping of rows and columns based on functional information in the original figure is consistent with the clustering pattern observed in the heatmap (Figure S5). As the original figure is more informative and better structured, we have included the new figure in the supplementary materials.

No statistical tests are provided for Figure 4.

We now indicate statistical significance in the figure and describe the statistical analysis in the figure legend, as suggested. Additionally, Table S5 is dedicated to the statistical analysis of the dNTP data.

**Reviewer #2 (Public Review):**
In this study, the authors assess whether selective pressure from drug chemotherapy influences the emergence of drug resistance through the acquisition of genetic mutations or phenotypic tolerance. I commend the authors on their approach of utilizing the mutation accumulation (MA) assay as a means to answer this and whole genome sequencing of clones from the assay convincingly demonstrates low mutation rates in Mycobacteria when exposed to sub-inhibitory concentrations of antibiotics. Also, quantitative PCR highlighted the upregulation of DNA repair genes in Mycobacteria following drug treatment, implying the preservation of genomic integrity via specific repair pathways.Even though the findings stem from M. smegmatis exposure to antibiotics under in vitro conditions, this is still relevant in the context of the development of drug resistance so I can see where the authors' train of thought was heading in exploring this. However, I think important experiments to perform to more fully support the conclusion that resistance is largely associated with phenotypic rather than genetic factors would have been to either sequence clones from the ciprofloxacin tolerance assay (to show absence/ minimal genetic mutations) or to have tested the MIC of clones from the MA assay (to show an increase in MIC).

Thank you for acknowledging the values of the manuscript and for the insightful suggestions for improvement. We agree on the necessity to directly connect the mutation accumulation experiments with the tolerance assay, and we have performed both suggested additional experiments.

(1) We repeated the ciprofloxacin tolerance assay (Figure S6) using a large number of plates to gather enough cells for genomic DNA extraction and whole genome sequencing. The sequencing confirmed the absence of mutations in bacteria grown in both 0.3 and 0.5 ug/ml ciprofloxacin. We integrated this result in the revised manuscript text, while the sequencing data are available at the European Nucleotide Archive (ENA) with PRJEB71590 project number.

(2) We resuscitated three different clones from the MA assays stored at -80°C and tested the MIC of the respective drugs. The results are presented in Figure 2C. Except for EMB, we observed an increase in MIC values across the treatments.

There seems to be a disconnect between making these conclusions from experiments conducted under different conditions, or perhaps the authors can clarify why this was done.

Molecular biology analysis methods are not easily compatible with long-term mutation accumulation experiments, or at least we could not establish the necessary conditions. When DNA or RNA extraction was required, we had to adjust the experimental scale for further analysis, which could be done in liquid culture. We believe that the suggested critical back-and-forth control experiments have significantly improved the comparability of the results.

With regards to the sub-inhibitory drug concentration applied, there is significant variation in the viability as calculated by CFUs following the different treatments and there is evidence that cell death greatly affects the calculation of mutation rate (PMCID: PMC5966242). For instance, the COMBO treatment led to 6% viability whilst the INH treatment led to 80% cell viability. Are there any adjustments made to take this into account?

We agree with and have been aware of the notion that cell death affects the calculation of the mutation rate. We included treatment optimization data on agar plates (Table 1 and Figure S2), which now demonstrate that the applied subinhibitory drug concentrations resulted in ≤10% viability across all treatments in the MA assay. This minimizes the potential discrepancy in the mutation rate calculation caused by variable cell death.

It would also be useful to the reader to include a supplementary table of the SNPs detected from the lineages of each treatment - to determine if at any point rifampicin treatment led to mutations in rpoB, isoniazid to katG mutations, etc.Overall, while this study is tantalizingly suggestive of phenotypic tolerance playing a leading role in drug resistance (and perhaps genetic mutations a sub-ordinate role) a more substantial link is needed to clarify this.

The SNPs identified from the lineages of each treatment are compiled in the 'unique_muts.xls' file within the Figshare document bundle that was originally enclosed with the manuscript. In response to your suggestion, we have now added a simplified version of this data set in Table S2, listing the detected SNPs. Notably, no confirmed adaptive mutation developed in our experiments; rifampicin treatment did not result in mutations in *rpoB*, nor did isoniazid lead to mutations in *katG*.

**Recommendations For The Authors:**
I would suggest moving Figure 1 to the supplementary - it shows that cell wall targeting drugs cause cell shortening and DNA replication targeting drugs cause cell elongation as would be expected and this is simply a secondary observation, not one that is central to the paper.

We agree that this is not a novel or unexpected observation. However, we used it as an indicator of drug effectiveness, particularly for bacteriostatic cell wall-targeting drugs in liquid culture that induced moderate cell death. Following Reviewer 1's suggestions, we extensively revised the figure to better convey our intended message. We believe the updated version now more clearly demonstrates the drugs' impact, and for this reason, we have opted to keep it in the main text.

Figure 2 and Table 2 show the same data so this can be combined as a paneled figure or one moved to the supplementary. It would be useful to include a diagram of how the MA assay was conducted, similar to the CIP tolerance assay figure.

Thank you for the suggestions. We have added a diagram to Figure 2 explaining the MA assay (Figure 2A), as well as the MIC experiment conducted on the MA cells (Figure 2C). To avoid redundancy, Table 2 has been removed.

**Reviewer #3 (Public Review):**
Summary:This manuscript describes how antibiotics influence genetic stability and survival in Mycobacterium smegmatis. Prolonged treatment with first-line antibiotics did not significantly impact mutation rates. Instead, adaptation to these drugs appears to be mediated by upregulation of DNA repair enzymes. While this study offers robust data, findings remain correlative and fall short of providing mechanistic insights.Strengths:The strength of this study is the use of genome-wide approaches to address the specific question of whether or not mycobacteria induce mutagenic potential upon antibiotic exposure.Weaknesses:The authors suggest that the upregulation of DNA repair enzymes ensures a low mutation rate under drug pressure. However, this suggestion is based on correlative data, and there is no mechanistic validation of their speculations in this study.Furthermore, as detailed below, some of the statements made by the authors are not substantiated by the data presented in the manuscript.Finally, some clarifications are needed for the methodologies employed in this study. Most importantly, reduced colony growth should be demonstrated on agar plates to indicate that the drug concentrations calculated from liquid culture growth can be applied to agar surface growth. Without such validations, the lack of induced mutation could simply be due to the fact that the drug concentrations used in this study were insufficient.

Thank you for appreciating the manuscript's merits and for the instructive suggestions. We agree that demonstrating reduced colony growth on agar plates is important to validate the relevance of the drug concentrations used in the study. In response, we have added the treatment optimization data on agar plates in Figure S2 and reorganized Table 1 to show the decrease in CFU achieved with the applied subinhibitory drug concentrations.

We acknowledge that the observed upregulation of DNA repair enzymes and the low mutation rates under drug pressure represent correlative data. We removed the reference to mechanism from the abstract and avoided presenting the qPCR results as a mechanistic explanation in the text. We have only raised the possibility that correlation could be a causal relationship: "The observed upregulation of the relevant DNA repair enzymes might account for the low mutation rate even under drug pressure." We recognize the necessity for a new series of targeted experiments to provide mechanistic explanations. We added the following text to the Discussion:

“The observed activation of DNA repair processes likely mitigates mutation pressure, ensuring genome stability. However, to confirm this hypothesis, these investigations should be conducted using genetically modified DNA repair mutant strains.”

In the current manuscript, we aim to convincingly demonstrate that long-term antibiotic pressure did not induce the occurrence of new adaptive mutations.

**Recommendations For The Authors:**
Additional specific comments are:Page 2. Do not italicize "Mycobacteria", which is not considered a scientific name.

Corrected.

Page 4. "Bacto pepcone" is a typo.

Corrected.

Page 6. "Quiagen" is a typo.

Corrected.

Page 9. In Table 1, RIF being described as a protein synthesis inhibitor is misleading.

Corrected.

Page 9. The statement "Specifically, following RIF, CIP, and MMC treatments, we observed cells elongating by more than twofold, whereas INH and EMB treatments led to a reduction in cell length." cannot be justified by Figure 1, as the cell length information is not conveyed in this figure.

Thank you for pointing this out, the revised Figure 1 conveys the cell length information.

Page 10. If the experiment shown in Figure S1 was done in an acidic growth condition, the figure legend should clearly indicate the fact. Additionally, the assay condition should be described in detail in the Methods section.

Thank you, the required information is now included in both the figure legend and the Methods section.

Page 10. If PZA does not work against M. smegmatis, it seems pointless to add it to the COMBO treatment. Please clarify why it was included in the drug combination experiment.

We added the following text to clarify the use of PZA: “Regardless of its inefficacy as a monotherapy, we included PZA in the combination treatment, as we could not rule out the possibility that PZA interacts with the other three drugs or that PZA elimination mechanisms are equally active in *M. smegmatis* under this regimen.”

Page 10. Generation times calculated from liquid culture cannot be applied to colony growth on an agar plate. The growth behaviors on a solid surface will be totally different from planktonic suspension growth. The numbers of generations indicated here will be inaccurate.

You are absolutely right. We conducted an experiment to calculate the number of generations on plates under the same conditions as used in the MA assay. We found, indeed, a different (doubled) generation time from what was determined in liquid culture. We have adjusted the mutation rates accordingly.

Page 12. Was the experiment shown in Figure 3 done in a liquid culture? If so, the transcriptional profile could be different from the experiment shown in Figure 2, which was done on an agar plate.

Yes, the experiment shown in Figure 3 was conducted in liquid culture. We acknowledge that the transcriptional profile could differ from the experiment shown in Figure 2, which was performed on an agar plate. However, technical limitations required us to use liquid cultures for these experiments.

Page 14. Regarding the statement "INH and EMB coincided with a decreased concentration of these [dCTP and dTTP] nucleotides", by examining Table S5, I do not see any statistical reductions in dCTP and dTTP levels.

Thank you for bringing this to our attention. We have made the necessary corrections to ensure that the text and data are now aligned.

Page 14. Similarly to the comment above, the statement "RIF, CIP and MMC treatments promoted an increase in the dCTP and dTTP pools" is misleading as each drug seems to increase either dCTP or dTTP, not both.

Same as above.

Page 14. The authors state, "a larger overall dNTP pool size coincides with a larger cell size and vice versa (Figure 4H)". Please indicate the unit of the pool size for the graph shown in Figure 4H. According to the legend, I assume that it refers to the concentration. The term "pool size" may be misleading as it implies quantity rather than concentration.Page 15. Figure 4H is impossible to understand. The left y-axis label looks as if it is a ratio of cell length to volume. There is no point in having these three data on a single graph. Please separate them into individual graphs. Also, what is the spacing between the tick marks? The data also seem inconsistent with the values given in Table S1. For example, the mean volume of COMBO is larger than the control (according to Table S1), and yet the graph in Figure 4H indicates that COMBO's relative length is less than 1.

Thank you for your feedback. We have corrected these and created what we hope is a clearer figure.

Figure S1. Clarify what the gray shade in the graph represents.

The gray shade was unnecessary, so we removed it when recoloring the figure to ensure a more coherent color scheme across the different treatments.

Figure S1. Relative viability cannot be determined by OD600. CFU needs to be determined to assess cell viability.

Thank you. We changed the incorrect term viability to growth inhibition.